# Comprehensive uncertainty estimation of the timing of Greenland warmings of the Greenland Ice core records

Eirik Myrvoll-Nilsen[1], Keno Riechers[1,2], Martin Rypdal[3], and Niklas Boers[1,2,4]

[1]Potsdam Institute for Climate Impact Research, Potsdam, Germany
[2]Technical University of Munich, Germany; School of Engineering & Design, Earth System Modelling
[3]The University of Tromsø – The Arctic University of Norway, Norway
[4]Department of Mathematics and Global Systems Institute, University of Exeter, UK

**Correspondence:** Eirik Myrvoll-Nilsen (myrvoll@pik-potsdam.de)

**Abstract.** Paleoclimate proxy records have non-negligible uncertainties that arise from both the proxy measurement and the dating processes. Knowledge of the dating uncertainties is important for a rigorous propagation to further analyses; for example for identification and dating of stadial-interstadial transitions in Greenland ice core records during glacial intervals, for comparing the variability in different proxy archives, and for model-data comparisons in general. In this study we develop a statistical

framework to quantify and propagate dating uncertainties in layer-counted proxy archives using the example of the Greenland Ice Core Chronology 2005 (GICC05). We express the number of layers per depth interval as the sum of a structured component that represents both underlying physical processes and biases in layer counting, described by a regression model, and a noise component that represents the fluctuations of the underlying physical processes, as well as unbiased counting errors. The joint dating uncertainties for all depths can then be described by a multivariate Gaussian process from which the chronology (such

as the GICC05) can be sampled. We show how the effect of a potential counting bias can be incorporated in our framework. Furthermore we present refined estimates of the occurrence times of Dansgaard-Oeschger events evidenced in Greenland ice cores together with a complete uncertainty quantification of these timings.

## 1 Introduction

The study of past climates is based on proxy measurements obtained from natural climate archives such as cave speleothems,

lake and ocean sediments, and ice cores. Paleoclimate reconstructions derived from proxies suffer from threefold uncertainty. First, the proxy measurement itself involves the typical measurement uncertainties. Second, the interpretation of proxy variables such as isotope ratios in terms of physical variables such as temperature is often ambiguous, and typically no one-to-one mapping can be established between the measured proxies and the climatic quantities of interest. Third, the age has to be measured alongside the proxy variable. In most cases an age model can be inferred, that provides a quantitative relationship between

the depth in the archive under consideration and the corresponding age. Such age models are also subject to uncertainties.

     This study is exclusively concerned with the dating uncertainties of so-called layer counted archives, where the dating is performed based on counting periodic signals in the proxy archives such as annual layers arising from the impact of the seasonal cycle on the deposition process (e.g., Rasmussen et al., 2006). This type of archives comprises varved lake sediments,

ice cores, banded corals, tree rings and some speleothems (Comboul et al., 2014). Using the example of the NGRIP ice
core (North Greenland Ice Core Project members, 2004) and its associated chronology – the GICC05 (Vinther et al., 2006;
Rasmussen et al., 2006; Andersen et al., 2006; Svensson et al., 2008) – , we present here a statistical approach to generate
ensembles of age models that may in turn be used to propagate the age uncertainties to any subsequent analysis of the time
series derived from the NGRIP record. Our method can be directly adapted to other layer-counted archives.

Layer counting assesses the age increments along the axis perpendicular to the layering, whose summation yields the total
age. In turn, also the errors made in the counting process accumulate such that in chronologies obtained from counting annual
layers, the absolute age uncertainty grows with increasing age (e.g. Boers et al., 2017).

Most importantly, dating uncertainties make it challenging to establish an unambiguous temporal relation between signals
recorded in different, and possibly remote, archives. Therefore, it is often not possible to decipher the exact temporal order
of events and distinguish causes from consequences across past climate changes. For example, abrupt Greenland warmings
known as Dansgaard-Oeschger (DO) events (Dansgaard et al., 1993; Johnsen et al., 1992) evidenced in ice core records from
the last glacial are accompanied by changes in the east Asian monsoon system, which are apparent from Chinese speleothem
records (e.g., Zhou et al., 2014; Li et al., 2017). However, since the dating uncertainties exceed the relevant time scales of these
abrupt climate shifts, a clear order of events cannot be determined. This prevents to deduce if in the context of DO events, the
abrupt Greenland warming triggered a hemispheric transition in the atmosphere, or vice versa, or if these changes happened
simultaneously as part of a global abrupt climatic shift (Corrick et al., 2020).

For the quantification of dating uncertainties in radiometrically dated archives, there exist well established generalized
frameworks. One example is the Bayesian Accumulation Model (Blaauw and Christeny, 2011) which models the sediment
accumulation rate as a first order autoregressive process with gamma distributed innovations. Other methods or software include
OxCal (Ramsey, 1995, 2008) and BChron (Haslett and Parnell, 2008; Parnell et al., 2008). Contrarily, the uncertainties of layer
counted archives are targeted systematically only by few studies. Comboul et al. (2014) present a probabilistic model, where
the number of missed and double-counted layers are expressed as counting processes shaped by corresponding error rates.
However, this approach requires knowledge about these rates and further does not account for any uncertainty associated with
them. An alternative Bayesian approach for quantifying the dating uncertainty of layer counted archives is presented in (Boers
et al., 2017), where the uncertainty is shifted from the time axis to the proxy value. However, this approach does not allow for
generation of ensembles of chronologies as required for uncertainty propagation.

Even though dating uncertainties are conveniently quantified for many archives, many studies ignore these and instead draw
inference from 'average' or 'most likely' age scales, as already highlighted by McKay et al. (2021). This involves the risk
of loosing valuable information, as for example shown in Riechers and Boers (2020). In some cases, rigorous propagation of
uncertainty may yield results that qualitatively differ from results obtained by using the 'average' or 'best fit' age model. In
this context, McKay et al. (2021) propose to apply the respective analysis to an ensemble of possible age scales to ensure the
uncertainty propagation, in line with the strategy proposed by Riechers and Boers (2020).

We focus on the layer counted part of the GICC05 chronology, a synchronized age scale for several Greenland ice cores
(Vinther et al., 2006; Rasmussen et al., 2006; Andersen et al., 2006; Svensson et al., 2008). It was obtained by counting the

layers of different Greenland ice cores and synchronizing the results using matchpoints. While the recent part of the chronology
is compiled from multiple cores, the older part (older than 15 kyr b2k) is based exclusively on the layer counting in the NGRIP
core. We introduce a new method to generate realistic age ensembles for the NGRIP core, which conveniently represent the
uncertainty associated with the GICC05.

Originally, the dating uncertainty of the GICC05 is quantified in terms of the the maximum counting error (MCE). The MCE
increases by 0.5 years for every layer which is deemed uncertain by the investigators during the counting process:

$$\text{MCE}(z) = 0.5 N_u(z), \tag{1}$$

where $N_u$ is the number of uncertain layers down to depth $z$. In contrast to certain layers, which can be identified unambigu-
ously in the records, uncertain layers are less pronounced and therefore it seems less certain that these signals truly correspond
to physical layers. The accumulation of uncertain layers results in high values for the MCE for the older parts of the core
(MCE = 2.6 kyr at 60 kyr b2k estimated age). However, it seems highly unlikely that all uncertain layers are consistently either
true layers or no layers, which is why we think that the MCE is an overly careful quantification of the age uncertainty, as
already suggested by Andersen et al. (2006). One might, alternatively, be tempted to treat the uncertain layers as a Bernoulli
experiment with $N_u$ repetitions and a probability of one half for each uncertain layer to be a true layer. However, this would
neglect any sort of bias in the assessment of the uncertain layers and would lead to unrealistically small uncertainties, since
over- and undercounting practically cancel each other in this Bernoulli type interpretation (see for instance Andersen et al.
(2006); Rasmussen et al. (2006)).

The method presented here abandons the notion of certain and uncertain layers. Instead, we separate the GICC05 chronology
into contributions that can be captured by deterministic model equations and corresponding residuals. We construct a new age-
depth model by complementing the deterministic part with a stochastic component designed in accordance with the statistics
of the residuals. This model can be used to generate age-depth ensembles in a computationally efficient manner. In turn,
these ensembles facilitate uncertainty propagation to subsequent analysis. The model parameters are tuned with respect to the
GICC05 chronology that includes every uncertain layer as half a layer.

The outline of this paper is as follows. Section 2 gives a description of the data used for this study. Section 3.1 introduces
our statistical model for the dating uncertainties, and details how we incorporate physical processes and how we deduce the
noise of the model from the statistics of the residuals. In section 3.2 we show how we can formulate our model in terms of a
hierarchical Bayesian modeling framework that allows for the physical and noise components to be estimated simultaneously.
This section also details how one can use the resulting posterior distributions of the model parameters to obtain a full description
of the posterior distributions of the dating uncertainties using a sample-based approach. We finally demonstrate how a potential
counting bias could be incorporated by the model, and how that would affect the results. In section 4 we show how our model
can be used to obtain a full description of the dating uncertainties of abrupt warming events, which takes into account the
dating uncertainties as well as the uncertainties in determining their exact position in the noisy data. Further discussion and
conclusions are provided in section 6.

## 2 NGRIP ice core data

We use the Greenland Ice core Chronology 2005 (GICC05) (Vinther et al., 2006; Rasmussen et al., 2006; Andersen et al., 2006; Svensson et al., 2008) as defined for the NGRIP ice core together with the corresponding $\delta^{18}$O proxy record (North Greenland Ice Core Project members, 2004; Gkinis et al., 2014; Ruth et al., 2003). An analogous analysis for the $Ca^{2+}$ proxy record is presented in Appendix B. The final age of the layer counted part of the GICC05 is 59,944 yr b2k, and we consider data up to 11,703 yr b2k. The following Holocene part of the record is excluded since it is governed by a substantially different climate than the last glacial interval (Rasmussen et al., 2014). For the considered period, the NGRIP record is available at 5 cm resolution and thus equidistant in depth, but not in time. In total, the data comprises $n = 18,672$ data points of the form $(z_k, y_k, x_k), k \in \{0, 1, ..., n-1\}$, where $z_k$ denotes the $k$th depth, $y_k$ the corresponding age as indicated by the GICC05, and $x_k$ the measured proxy value.

The GICC05 is based on counting annual layers which are evident in multi-proxy continuous flow measurements from the NGRIP, DYE3 and the GRIP ice cores. While the measurements from DYE3 and GRIP only facilitate layer counting up to ages of 8.2 kyr b2k and 14.9 kyr b2k, respectively, the NGRIP core allowed the identification of annual layers up to an age of 60 kyr b2k. The uncertainty of the GICC05 has been quantified as follows: whenever the investigators were uncertain about whether or not a signal in the data should be considered an annual layer, half a year was added to the cumulative number of layers while simultaneously adding $\pm 0.5$ to the age uncertainty. The total age uncertainty determined by the number of all uncertain layers up to a given depth is termed the maximum counting error (MCE). The MCE amounts to a relative age uncertainty of 0.84% at the onset of the Holocene and 4.34% at the end of the layer counted section of the core.

$\delta^{18}$O values from Greenland ice cores are interpreted as a qualitative measure of the site temperature at the time of precipitation (Jouzel et al., 1997; Gkinis et al., 2014). We include this data in our study since our modelling approach will make use of the relation between atmospheric temperatures and the amount of precipitation, which in turn affects the thickness of the annual layers. In addition we use the division of the record into Greenland stadial and interstadial phases as presented in Tab. 2 of Rasmussen et al. (2014). We label the depths at which stadial-interstadial transitions occur by $z_1^*, ..., z_p^*$ and the corresponding ages by $y_1^*, ..., y_p^*$. Figure 1 shows the measured $\delta^{18}$O values as a function of the GICC05 time scale, together with the Greenland stadial and interstadial onsets.

## 3 Methods

### 3.1 Age-depth model

We assume that depths $\boldsymbol{z} = (z_1, ..., z_n)^\top$ and proxy values $\boldsymbol{x} = (x_1, ..., x_n)^\top$ are measured accurately and hence treat them as deterministic variables. In contrast, we consider the ages $\boldsymbol{y} = (y_1, ...y_n)^\top$ as dependent stochastic variables and will in the following establish a model to map the independent depths and stable isotope concentrations onto ages, in a way that reflects the uncertainties inherent to the dating. The model will be supplemented with information on the prevailing climate period.

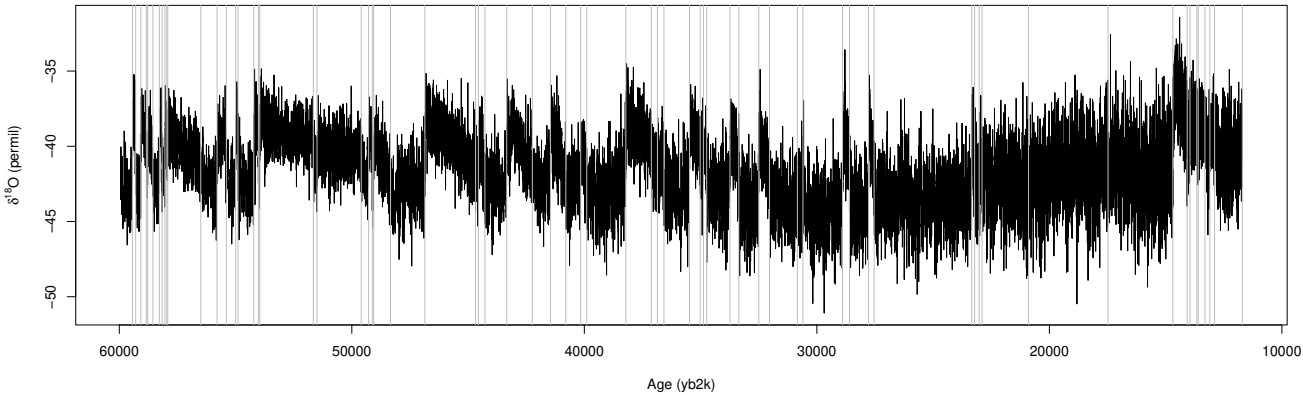

**Figure 1.** The measured $\delta^{18}$O isotope values plotted against the corresponding GICC05 time scale, starting from 11,793 yr b2k. The gray vertical lines denote the transitions between Greenland stadial and interstadial periods as reported by Rasmussen et al. (2014).

In order to motivate our modelling approach we give some general considerations about the deposition process as well as the counting process. The decisive quantity for us will be the incremental number of annual layers counted in a 5cm depth increment of the ice core

$$\Delta y_k = y_k - y_{k-1}. \tag{2}$$

This quantity is determined by the amount of precipitation (minus the snow that is blown away by winds) during the corresponding period, the thinning that the layers experience over time deeper down in the core and potential errors made during the counting process. While the thinning process can be expected to happen mostly deterministically, the net annual accumulation of snow certainly exhibits stronger fluctuations. Finally, the counting error adds additional randomness. Thus, it is reasonable to regard the observed age increments $\Delta\boldsymbol{y}$ as a realization of a random vector $\Delta\boldsymbol{Y}$ which can be decomposed into a deterministic and a stochastic component

$$\Delta Y_k = a(z_k) + \varepsilon_k. \tag{3}$$

Note that the number of layers within a 5 cm depth increment is not necessarily an integer number. Given that the amount of precipitation co-varies with atmospheric site temperatures we can specify

$$a(z_k) = a(z_k, x(z_k)) \rightarrow a(z_k, x_k). \tag{4}$$

Based on physical arguments and the analysis of the observed age increments $\Delta y_k$, we will propose the structural form of the deterministic part of the model and then tune the model parameters to the data. In turn, this allows us to design the model's noise component $\varepsilon$ in accordance with the corresponding residuals $\delta_k = \Delta y_k - a(z_k, x_k)$.

### 3.1.1 Linear Regression

As explained above, the thickness of the counted layers, and thereby the number of layers per depth increment $\Delta z_k = z_k - z_{k-1}$, is governed by two physical factors: the amount of precipitation at the time the layer was formed and the thinning of the core due to ice flow. These processes are here assumed to follow a regression model. We take into account the thinning by implementing a second order polynomial dependency of $\Delta Y_k$ with respect to the depth $z_k$. Choosing this nonlinear function conveniently accounts for the saturation of the layer thinning evident in the NGRIP ice core. The amount of precipitation is known to co-vary with atmospheric temperatures, since by the Clausius-Clapeyron relation the moisture holding capacity of the atmosphere increases with temperatures. This is represented using a linear response to the $\delta^{18}O$ measurements. The same response is applied to the $\log(Ca^{2+})$ in the alternative analysis presented in Appendix B. Finally, we observe clear trends in the incremental layers that persist over individual stadials and interstadials. Given the consistency of these trends across the different climate periods, we decided to incorporate them in the deterministic model component. Overall, we propose a deterministic model of the form

$$a(z_k, x_k) = b z_k^2 + b_x x_k(z_k) + \sum_{i=1}^{p} \psi_i(z_k; a_i, c_i), \tag{5}$$

with

$$\psi_i(z_k; a_i, c_i) = \begin{cases} a_i + c_i z_k, & z_i^* < z_k < z_{i+1}^* \\ 0, & \text{otherwise} \end{cases}, \tag{6}$$

in order to capture the systematic features of the chronology. Here, $c_i$ denote the period specific slopes and $a_i$ their corresponding offsets. For $p$ transitions between stadials and interstadials we have to tune $2p + 2$ regression parameters, which is achieved by fitting the above model for $a(z_k, x_k)$ to the observed layer increments $\Delta y_k$ given by the GICC05 time scale in a least square approach. As explained above, the GICC05 ages contain the contribution of uncertain layers, which were counted as half a year each. Here, we abandon the distinction of certain and uncertain layers and regard the GICC05 ages as the best possible estimate of the true ages and accordingly use them directly for the optimization. The fitted model is shown in red in the top panel of Fig. 2.

### 3.1.2 Noise structure

After tuning the deterministic part of Eq. 3, the residuals are given by

$$\delta_k = a(z_k, x_k) - \Delta y_k. \tag{7}$$

We find the residuals to be symmetric and unimodally distributed, and apart from some degree of over-dispersion they appear to be well described by a normal distribution, as shown in Fig. 3. Moreover, by examining the empirical autocorrelation illustrated in Figure 3c, we observe that the residuals exhibit a fast decay of memory which is indicative of stationarity. This suggests that the noise can be expressed using a short-memory Gaussian stochastic process.

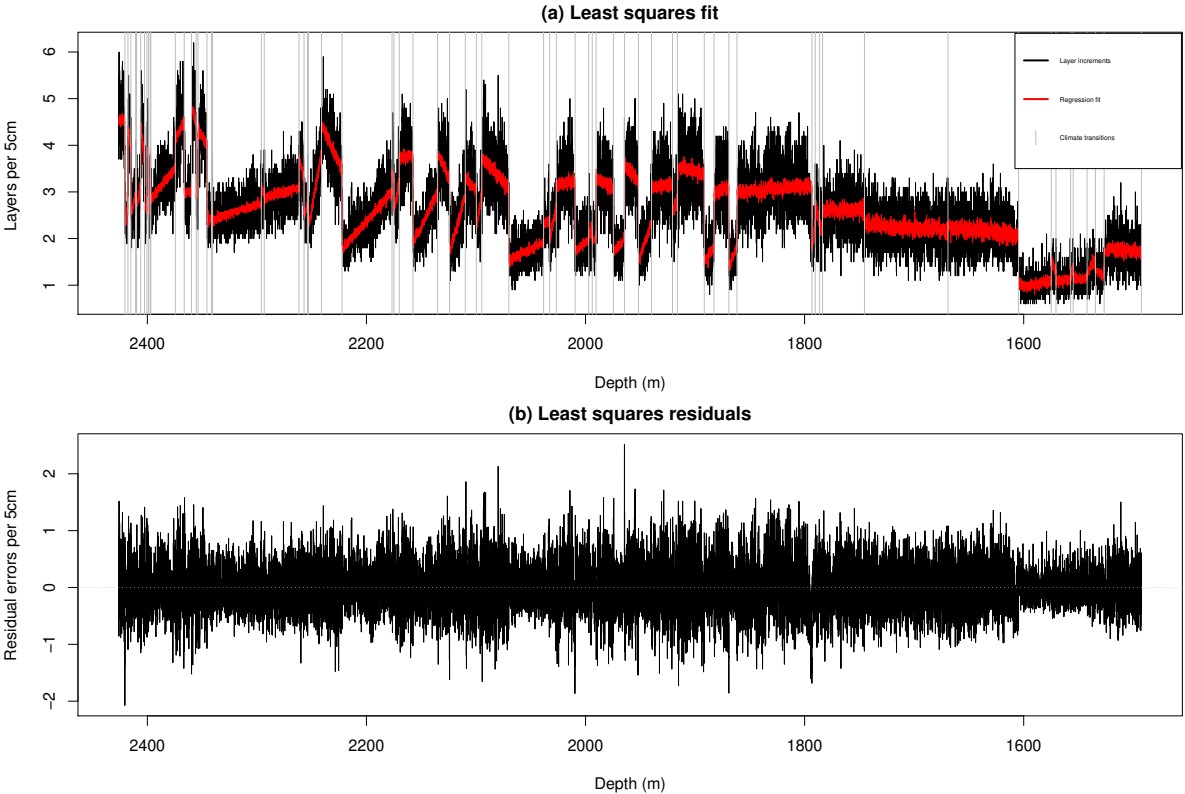

**Figure 2.** (a) Number of layers counted in the GICC05 time scale per 5cm depth increments in the NGRIP ice core (black). The red line shows the fitted values from the regression model $a(z_k, x_k) = bz_k^2 + b_x x(k) + \sum_{i=1}^{p} \psi_i(z_k; a_i, c_i)$. The gray vertical lines represent the transitions between Greenland stadials and interstadials. (b) The residuals $\delta_k$ obtained from fitting the regression model $a(z_k, x_k)$ to the layer increments $\Delta y_k$.

We explore three different models for the correlation structure of the noise $\varepsilon$. The first model assumes that they follow
independent and identically distributed (iid) Gaussian processes

$$\varepsilon_k \overset{iid}{\sim} \mathcal{N}(0, \sigma_\varepsilon^2). \tag{8}$$

The second model assumes that the noise can be described by a first-order autoregressive (AR) process

$$\varepsilon_k = \phi \varepsilon_{k-1} + \xi_k, \tag{9}$$

where $\phi$ is the first-lag autocorrelation coefficient and $\xi_k$ is a white noise process with variance $\sigma_\xi^2 = \sigma_\varepsilon^2/(1 - \phi^2)$. The third
model assumes the noise follows a second-order autoregressive process

$$\varepsilon_k = \phi_1 \varepsilon_{k-1} + \phi_2 \varepsilon_{k-2} + \xi_k, \tag{10}$$

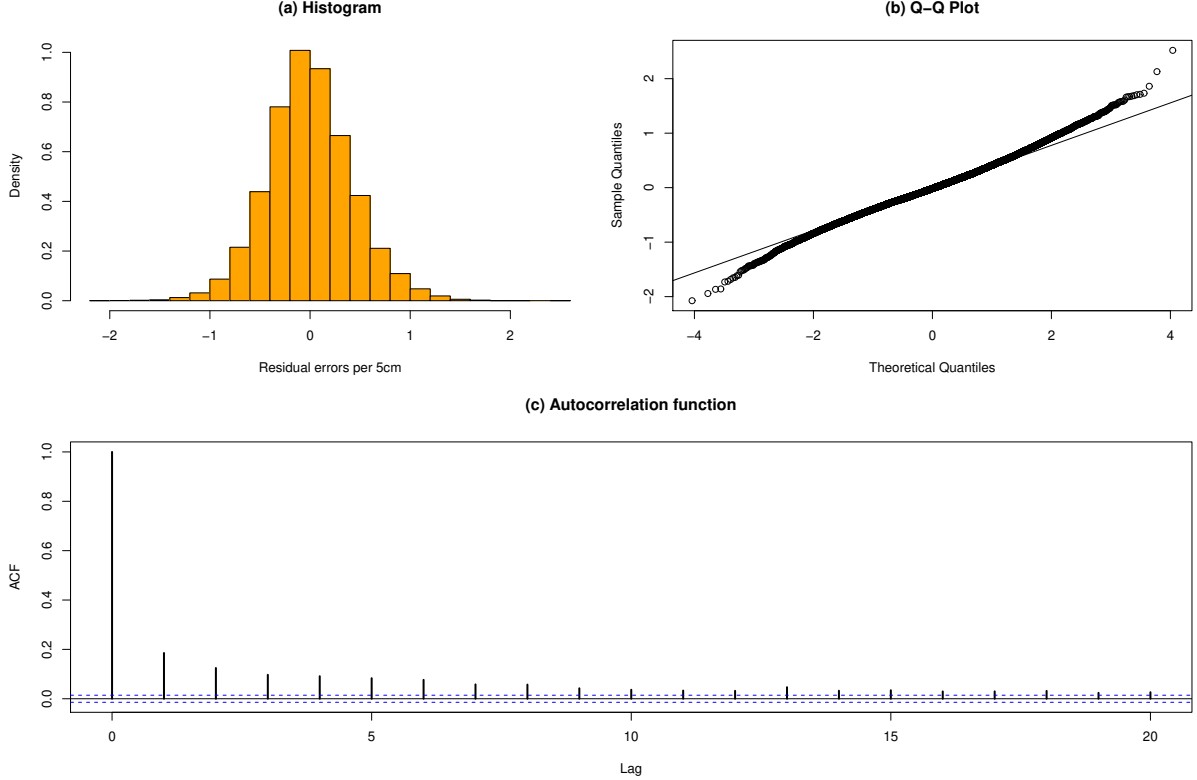

**Figure 3.** (a) Histogram of the residuals obtained from least squares fit of the regression model $a(z_k, x_k)$. (b) The corresponding quantile-quantile plot. We observe key properties of symmetry and unimodality indicating Gaussianity. The quantile-quantile plot indicates a small over-dispersion, but overall the data seem to be consistent with a normal distribution. (c) The autocorrelation function of the residuals from the least squares fit of the linear regression model, up to a maximum of 20 lags. A fast memory decay can be inferred.

where $\phi_1$ and $\phi_2$ are the first- and second-lag autocorrelation coefficients and $\xi_k$ is a white noise process with variance

$$\sigma_\xi^2 = \sigma_\varepsilon^2 \frac{1 - \phi_2}{(1 + \phi_2)\big((1 - \phi_2)^2 - \phi_1^2\big)}. \tag{11}$$

Note that a potential global or at least climate-regime-specific bias in the counting process, such as overseeing systematically
1 out of 10 layers, would be captured by the regression model and thus cannot be identified as a systematic error. We will
investigate the influence of potential systematic errors below. Similarly, fluctuations of the physical processes can be captured
by the noise model which aims to represent the counting errors. It would therefore be more accurate to interpret $a(z_k, x_k)$
as a structured component representing the part of both the physical processes and a systematic counting error which can be
accounted for by linear regression, and $\varepsilon_k$ as the fluctuations of both the physical processes and counting errors. Hence, $\varepsilon_k$ can
be considered an upper bound on the counting uncertainty.

## 3.2 Simultaneous Bayesian modeling

So far, we have fitted only the structural model component. This enabled us to investigate statistical properties of the residuals and formulate corresponding noise model candidates. Fitting both the linear regression model and the different noise models can in principle be performed in two stages: First, the linear regression model is fitted to the layer increments using the method of least squares. Thereafter, the fitted values are subtracted and the selected noise model is fitted to the residuals. However, this approach has the disadvantage that some variation that may in reality be caused by the noise process $\varepsilon_k$ may have been attributed to the structured component $a(z_k, x_k)$ and removed before fitting the noise model. We therefore introduce here a Bayesian approach that enables us to estimate all model parameters simultaneously. The Bayesian approach has three key advantages over the least square fitting of the structured component: First, it treats the noise and the structured component equally, and second, it returns the joint posterior probability of all model parameters which indicates the plausibility of a certain parameter configuration in view of the data. The posterior probability distribution can be regarded as an uncertainty quantification of the model's parameter configuration. Third, in the Bayesian parameter estimation, prior knowledge and constraints on the parameter can be incorporated via a convenient choice of the so-called prior distributions.

In general terms, let $\mathcal{D}$ denote some observational data and $\boldsymbol{\theta}$ denote parameters that shape a model which is assumed to reasonably describe the process that generated the data. Then Bayes' Theorem can be used to deduce the posterior probability density of the parameters $\boldsymbol{\theta}$ given the data $\mathcal{D}$:

$$\pi(\boldsymbol{\theta} \mid \mathcal{D}) = \frac{\pi(\mathcal{D} \mid \boldsymbol{\theta})\pi(\boldsymbol{\theta})}{\pi(\mathcal{D})}. \tag{12}$$

In our case the GICC05 age $\boldsymbol{y} = (y_1, ..., y_n)$, or more precisely their increments $\Delta \boldsymbol{y}$, represent the observational data $\mathcal{D}$ assumed to be generated from the model defined by Eq. 3. There are $2p + 2$ parameters $\boldsymbol{\beta} = (b_2, b_x, a_1, c_1, ..., a_p, c_p)$ for the structured component alone and the noise adds another 1 to 3 parameters, depending on the choice of the noise structure. Thus the set of model parameters reads

$$\boldsymbol{\theta} = (\boldsymbol{\beta}, \boldsymbol{\psi}), \tag{13}$$

where $\boldsymbol{\psi} = \sigma_\varepsilon$ if the residuals are assumed to follow an iid Gaussian distribution, $\boldsymbol{\psi} = (\sigma_\varepsilon, \phi)$ if they are assumed to follow an AR(1) process and $\boldsymbol{\psi} = (\sigma_\varepsilon, \phi_1, \phi_2)$ if they are assumed to follow an AR(2) process. For any given parameter configuration, the likelihood is for all three choices of the noise structure defined by a multivariate Gaussian distribution

$$\pi(\Delta \boldsymbol{y} \mid \boldsymbol{\theta}) = (2\pi)^{-n/2} |\boldsymbol{\Sigma}|^{-1} \exp\left\{ -\frac{1}{2}(\Delta \boldsymbol{y} - \boldsymbol{a})^\top \boldsymbol{\Sigma}^{-1}(\Delta \boldsymbol{y} - \boldsymbol{a}) \right\}, \tag{14}$$

where $\boldsymbol{a} = (a(z_1, x_1), ..., a(z_n, x_n))^\top$ and the entries of the autocovariance matrix $\boldsymbol{\Sigma}$ are given by the autocovariance function $\Sigma_{ij} = \gamma(|i - j|)$ of the assumed noise model. For the iid model, the autocovariance function is simply $\sigma_\varepsilon^2$ if $i = j$ and zero otherwise, resulting in a diagonal covariance matrix. For the AR(1) model the autocovariance function is

$$\gamma(k) = \frac{\sigma_\varepsilon^2}{1 - \phi^2} \phi^{|k|}. \tag{15}$$

The autocovariance function of the AR(2) model is specified by the difference equation

$$\gamma(k) = \phi_1 \gamma(k-1) + \phi_2 \gamma(k-2), \tag{16}$$

with initial conditions

$$\gamma(0) = \left(\frac{1-\phi_2}{1+\phi_2}\right)\frac{\sigma_\varepsilon^2}{(1-\phi_2)^2 - \phi_1^2} \tag{17}$$

$$\gamma(1) = \frac{\phi_1}{1-\phi_2}\gamma(0). \tag{18}$$

A benefit of having the likelihood follow a Gaussian distribution is that it can be evaluated easily and samples can be obtained efficiently, despite the large number of parameters. Finally, we define convenient priors for the model parameters. For the parameters of the structured model component $\boldsymbol{\beta}$ we choose vague Gaussian priors, with variances that safely cover all reasonable parameter configurations. For the noise parameters $\boldsymbol{\psi}$ we restrict the scaling parameter $\sigma_\varepsilon$ to be positive, and the autoregressive coefficients such that they define a stationary model. These constraints are embedded into the model by adopting suitable parametrizations. The scaling parameter $\sigma_\varepsilon$ is assigned a gamma distribution through the parametrization $\kappa = \log(1/\sigma_\varepsilon^2)$. For the lag-one correlation parameter in the AR(1) model we assume a Gaussian prior on the logit transformation $\rho = \log((1+\phi)/(1-\phi))$. For the AR(2) model we instead assign priors on the logit transformation of the partial autocorrelations $\psi_1 = \phi_1/(1-\phi_2)$ and $\psi_2 = \phi_2$, using penalised complexity priors (Simpson et al., 2017).

In principle, the joint posterior density can then be sampled from by using a Markov chain Monte Carlo (MCMC) algorithm (e.g., Goodman and Weare, 2010). However, to solve Eq. 12 more efficiently, we formulate the problem in terms of a latent Gaussian model and then use integrated nested Laplace approximations (INLAs) (Rue et al., 2009, 2017) to compute the joint and marginal posterior distributions (for details see Appendix).

The posterior distribution of $\boldsymbol{\theta}$ enables us to generate ensembles of different realizations of the random variable $\Delta \boldsymbol{Y}$, i.e., of the age increments that correspond to the fixed depth increments $\Delta \boldsymbol{z}$. In a two-stage Monte Carlo simulation, first a value for $\boldsymbol{\theta}$ is randomly sampled from the posterior $\pi(\boldsymbol{\theta} \mid \Delta \boldsymbol{y})$. Second, the noise $\boldsymbol{\varepsilon}$ is sampled according to the noise model using noise parameters sampled in the first step. An ensemble generated in this fashion simultaneously reflects the uncertainty enshrined in the stochastic process and the model thereof as well as the uncertainty about the model parameters. Each realization of age increments yields a corresponding possible chronology according to

$$y_k = y_0 + \sum_{i=1}^{k}\left(a(z_i, x_i) + \varepsilon_i\right), \tag{19}$$

where $y_0$ is the number of reported layers up to the depth $z_0$. Fig. 4 shows the 95% credible intervals obtained from age ensembles for the three different noise models with respect to the GICC05 age. Each ensemble comprises 10000 realizations of $\boldsymbol{Y}$. In this plot we notice a significant increase of uncertainty going from the iid to the AR(1) model. This is intuitive as when more memory is added to the model, the variation increases. However, going from AR(1) to AR(2) adds only moderate additional uncertainty. We therefore argue that an AR(1) process is sufficient in terms of modeling the correlation structure of the residuals. The same is observed when using $\log(Ca^{2+})$ as a proxy instead, see Fig. B1. All following computations are hence carried out with the AR(1) noise model.

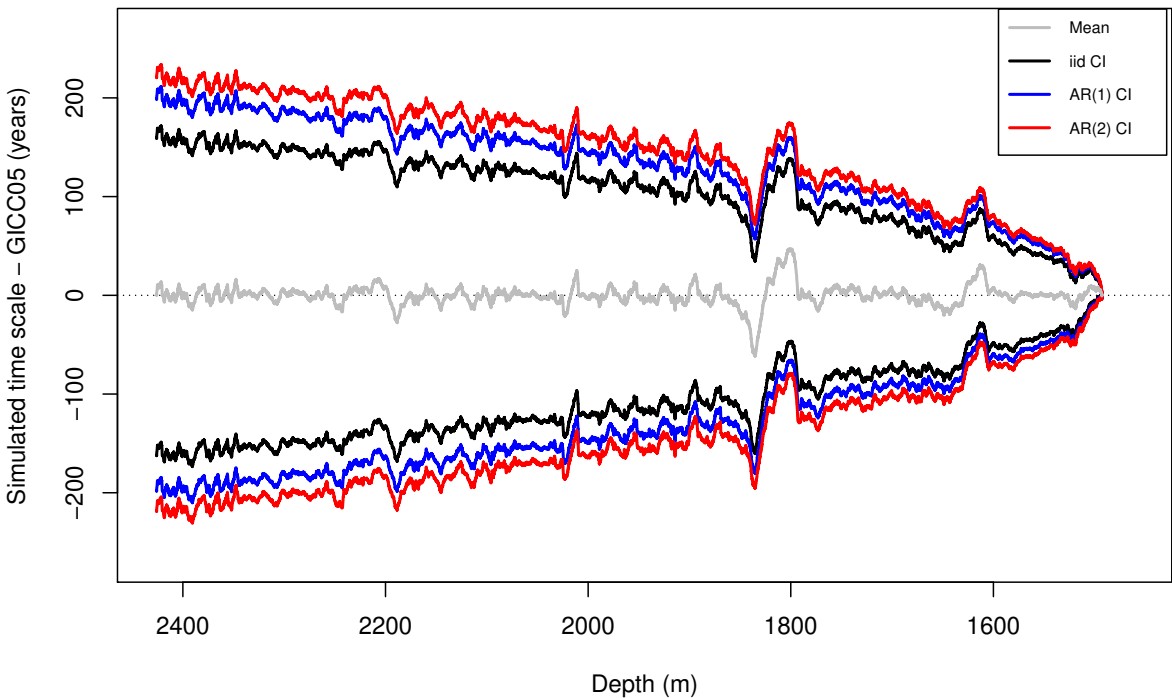

**Figure 4.** 95% credible intervals of the dating uncertainty distribution when $\delta^{18}O$ is used as the proxy covariate. The GICC05 time scale has been subtracted and the noise is modeled using iid (black), AR(1) (blue) and AR(2) (red) noise models. Only the posterior marginal mean computed using AR(1) distributed noise is included (gray) since it is very similar to the mean obtained using other noise assumptions.

### 3.3 Incorporating an unknown counting bias

When originally quantifying the uncertainty of the GICC05 chronology, a concern was that the layer counting was potentially
biased, in the sense that layers were consistently over-counted or missed. As highlighted by Andersen et al. (2006), there is no way to quantify a potential bias based on the data. Here, we investigate the influence that such a bias would have on our model, assuming a given maximum bias strength. To capture the effect of a systematic bias in the the layer counting we introduce a scaling parameter $\eta$ such that

$$\Delta Y_k = \eta \left( a(z_k, x_k) + \varepsilon_k \right). \tag{20}$$

Given that we have no knowledge about the size of the bias, $\eta$ must be regarded as a random variable, whose distribution can only be estimated by experts a priori. Originally, biases on the order of 1% in the counting performed by different investigators have been observed (Rasmussen et al., 2006; Andersen et al., 2006). Here we assume

$$\eta \sim \mathcal{U}(1 \pm \Delta\eta), \tag{21}$$

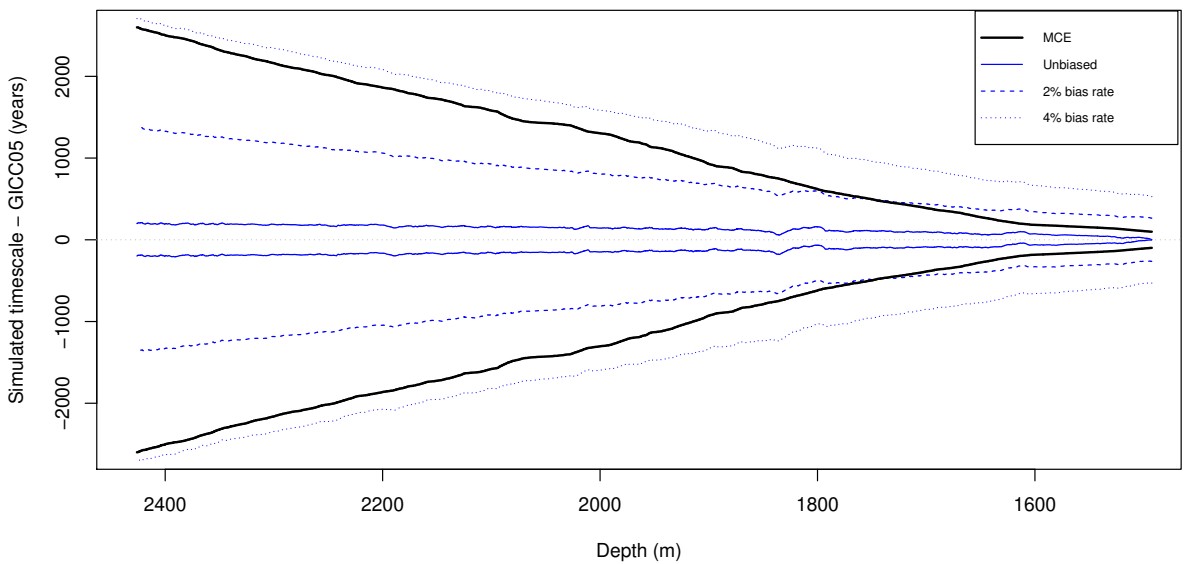

**Figure 5.** The 95% credible intervals of the difference between estimated dating and GICC05 time scale compared to the maximum counting error (solid black). Dating uncertainties in this case include a bias expressed by a stochastic scaling parameter drawn from a uniform $\mathcal{U}(1 \pm \Delta\eta)$ distribution. The solid blue line represents the unbiased $\Delta\eta = 0$ dating uncertainty, while the dashed and dotted blue lines represent the biased cases of $\Delta\eta = 2\%$ and $\Delta\eta = 4\%$, respectively. The differences in dating uncertainty between iid, AR(1) and AR(2) models are dwarfed by the uncertainty introduced by the unknown bias. Hence, only the AR(1) distributed residuals are shown here.

meaning the layer-counters are just as likely to systematically over-count as to under-count, on a maximum rate of $\Delta\eta$. While the expectation for the age increments $\mathrm{E}(\Delta Y)$ remains unchanged as long as $\mathrm{E}(\eta) = 1$, their variance will grow due to the additional uncertainty. Fig. 5 shows the 95% credible intervals for potentially biased chronology ensembles generated under the assumption that $\Delta\eta$ equals 0%, 2% or 4%.

It is evident that the bias uncertainty contributes substantially to the age uncertainty. This is expected since a relatively small counting bias yields a large absolute error at a possible age of 60 kyr b2k, which in turn exceeds the uncertainty contribution of the noise by far. We also observe that one would need a maximum error rate of $\Delta\eta \approx 4\%$ in order for the uncertainty to approach the maximum counting error at the end of the layer-counted core segment.

## 4 Examples of applications

### 4.1 Dating uncertainty of DO-events

Both the NGRIP $\delta^{18}O$ and $\log(Ca^{2+})$ records are characterized by prominent abrupt shifts from low to high values, including the so-called Dansgaard-Oeschger (DO) events (Johnsen et al., 1992; Dansgaard et al., 1993). These jumps are interpreted as

sudden warming events in Greenland, which took place repeatedly during the last glacial period. In order to explain the physical relationship between these abrupt Greenland warming events and apparently concomitant abrupt climate shifts evidenced in other archives from different parts of the planet, it is crucial to disentangle the exact temporal order of these events. This requires a rigorous treatment of the uncertainties associated with the dating of DO events in Greenland ice core records.

Rasmussen et al. (2014) provide a comprehensive list of Dansgaard-Oeschger events and other stadial-interstadial transitions, indicating depths from the NGRIP ice core at which they occur, and the corresponding GICC05 age. They report the visually identified event onsets, and provide uncertainty estimate in terms of data points along the depth axis and the respective MCE associated with the estimated event onset depth. This assessment was later refined by Capron et al. (2021), using the algorithm for detecting transition onsets designed by Erhardt et al. (2019). Here, we present a rigorous combination of the depth and age

uncertainties, which complicate the exact dating of abrupt warming events.

    First, we adopt the Bayesian transition onset detection designed by Erhardt et al. (2019) to estimate the onset of the abrupt warming transitions in the proxy records with respect to the depth in the core. By $Z^*$ we denote a continuous stochastic variable that represents the uncertain onset depth and by $x^*$ we denote a selected data window of the proxy record enclosing the transition. For each transition this yields a posterior distribution $\pi(Z^* \mid x^*)$ over potential transition onset depths, assuming

a linear transition from low to high proxy values perturbed by AR(1) noise. For inference we adopt the methodology of INLA as it is particularly suited for such models, granting us a significant reduction in computational cost over traditional MCMC algorithms. The application of the transition onset detection to the onset of GI-11 is presented in Fig. 6a, with the resulting posterior marginal distribution for $Z^*$ illustrated in Fig. 6b. Note that the dating method for the transitions is sensitive to the choice of the data window. In App. D we detail how the fitting windows for the linear ramps are optimally chosen. The selected

data windows are listed in Tab. D1. Furthermore, the Bayesian transition detection fails in some cases where the transition amplitudes are small. We successfully derive posterior marginal distribution for in total 29 events, whose summary statistics are listed in Tab. 1.

    Each potential transition onset depth $z^*$ yields a distribution over potential transition onset ages $Y^*$. This uncertainty, denoted by $\pi(Y^* \mid \Delta y, z^*)$, is determined by linearly interpolating the age ensemble members generated according to Sec. 3.1 based

on the observed layer increments $\Delta y$. The posterior distribution for the transition onset date for a given DO event thus reads

$$\pi(Y^* \mid \Delta y, x^*) = \int \pi(Y^* \mid \Delta y, Z^*)\pi(Z^* \mid x^*)dZ^*. \tag{22}$$

    Technically, we create an ensemble of potential onset ages in the following way. First, we generate an ensemble of 10,000 samples from the posterior distribution of the transition onset depth

$$z_r^* \sim \pi(Z^* \mid x^*), \quad r \in [1, 10000]$$

to represent the onset depth uncertainty. Second, for each onset depth sample $z_r^*$ we produce a simulation of a chronology from the corresponding age uncertainty

$$y_r^* \mid z_r^* \sim \pi(Y^* \mid \Delta y, z_r^*).$$

Thus, for both proxies ($\delta^{18}$O and $Ca^{2+}$) and for each event we obtain 10,000 possible values for the transition onset age whose distribution corresponds to the posterior distribution expressed in (22). The posterior marginal mean and 95% credible intervals for each event are reported in Tab. 1. This table hence gives the timing of the transitions together with the full uncertainties, stemming from the transition onset detection and the dating of the record. The estimated dating uncertainties are presented visually in Fig. 7, where they are also compared to the onset age obtained by Rasmussen et al. (2014), as well as the best estimates from Buizert et al. (2015) and Capron et al. (2021) which are presented in Tab. 2.

Although the GICC05 ages $y^*$ of abrupt warming events and subevents as reported by Rasmussen et al. (2014) fall within our estimated 95% credible intervals for all transitions, there are some transitions where there is a notable difference between the estimated posterior marginal mean $E(Y^*)$ and the reported $y^*$. Partially, this can be explained by the fact that Rasmussen et al. (2014) uses a lower 20-year temporal resolution, and that they determine the onset from three different ice cores and two different proxies ($\delta^{18}$O and $Ca^{2+}$), whereas our assessment is based on the univariate NGRIP proxy records only. With $E(Y^*) - y^* \sim 120$ the difference is most prominent in the GI-11 transition, whose simulated ages are represented in the histogram in Fig. 6c. This discrepancy is caused by the difference between our estimated onset depth $Z^*$ and $z^*$ from our linear ramp model fit shown in Fig. 6a. Our estimated onset depth $E(Z^*)$ differs from the value $z^*$ reported by Rasmussen et al. (2014) by approximately 1.6 m. This discrepancy propagates into an accordingly large difference in the age estimation of the transition onset. This demonstrates the importance of incorporating proper estimation and uncertainty quantification of the onset depth. Although the absolute uncertainty added from determining the onset depth can be considered negligible compared to the much larger age-depth uncertainty, there can still be a noticeable shift in the estimated onset age propagated from the estimation of the onset depth.

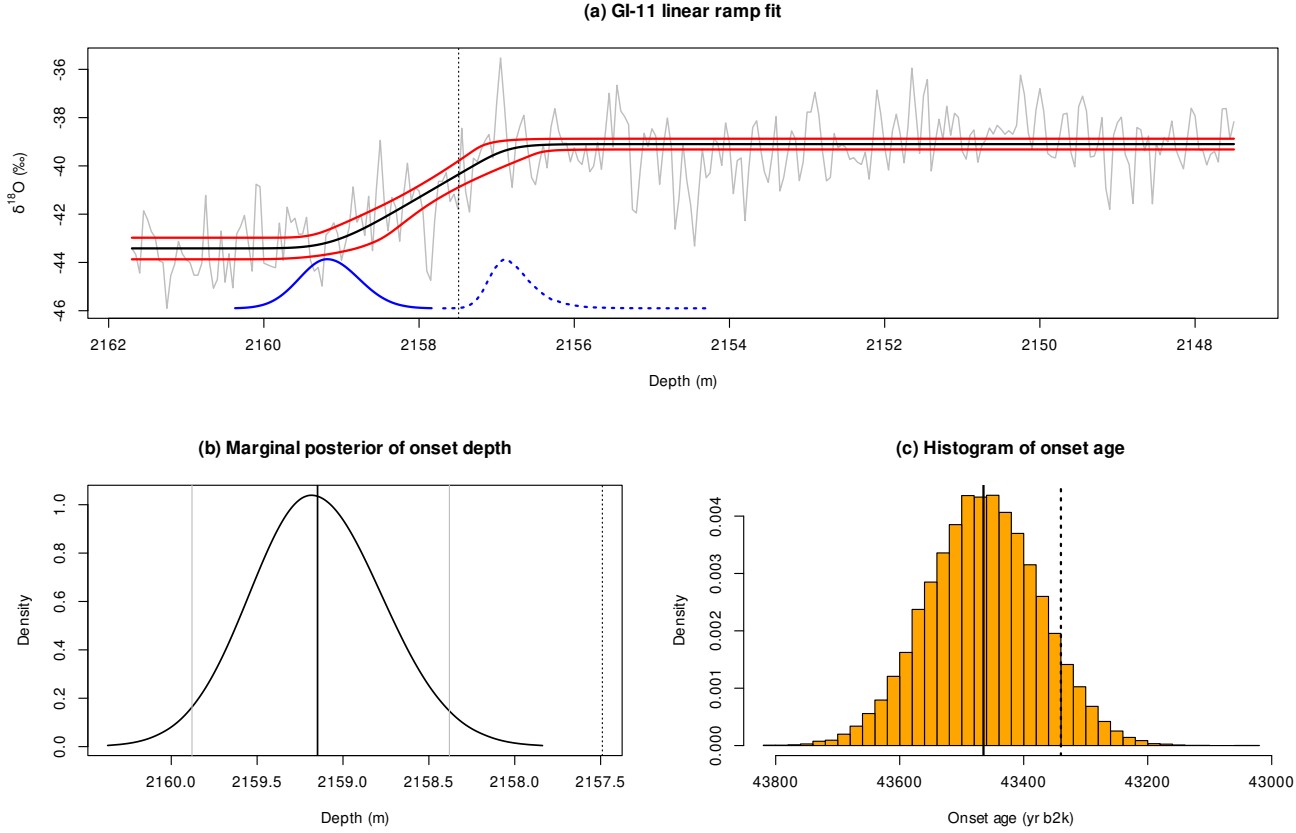

**Figure 6.** (a) The recorded $\delta^{18}O$ of the NGRIP data $\boldsymbol{x}^*$ (gray) as a function of depth for the GI-11 transition. The black line represent the posterior marginal mean of the linear ramp model fitted using INLA. The enclosing red lines represent the corresponding 95% credible intervals. The blue curves at the bottom illustrate the (unscaled) posterior distributions of the onset (solid) and end point (dotted) of the transition. The vertical dotted line describe the onset depth $z^*$ as reported in Rasmussen et al. (2014). (b) The posterior marginal distribution of the onset depth $Z^*$ following a linear ramp model fit. The solid vertical lines represent the posterior marginal mean (black) and 95% credible intervals (gray). The dotted vertical line represent $z^*$. (c) Histogram describing the complete dating uncertainty of the transition to GI-11, taking into account the uncertainty of the NGRIP depth of the onset as well as the dating uncertainty at this depth. The solid black vertical line represent the mean of these samples, $E(Y^*)$, and the dotted vertical line represent the GICC05 onset age $y^*$ as reported in Rasmussen et al. (2014).

| Event | $z^*$ (m) | $Z^*$ mean (m) | $Z^*$ 95% CI (m) | $y^*$ (yb2k) | $Y^*$ mean (yb2k) | $Y^*$ 95% CI (yb2k) |
|---|---|---|---|---|---|---|
| GI-1d | 1574.80 | 1574.97 | (1574.85, 1575.11) | 14075 | 14078.53 | (14020.02, 14136.57) |
| GI-1e | 1604.64 | 1604.52 | (1604.47, 1604.57) | 14692 | 14689.13 | (14621.21, 14758.25) |
| GI-2.2 | 1793.19 | 1793.87 | (1793.63, 1794.08) | 23340 | 23383.98 | (23269.8, 23497.12) |
| GI-3 | 1869.12 | 1869.22 | (1868.95, 1869.56) | 27780 | 27789.43 | (27661.18, 27916.42) |
| GI-4 | 1891.57 | 1891.67 | (1891.32, 1892.05) | 28900 | 28909.82 | (28777.57, 29042.34) |
| GI-5.2 | 1951.65 | 1952.02 | (1951.98, 1952.06) | 32500 | 32524.76 | (32383.73, 32662.2) |
| GI-6 | 1974.55 | 1974.40 | (1974.36, 1974.44) | 33740 | 33734.54 | (33590.28, 33874.96) |
| GI-7b | 1997.04 | 1997.25 | (1997.20, 1997.31) | 35020 | 35029.43 | (34881.88, 35173.95) |
| GI-7c | 2009.44 | 2009.76 | (2009.70, 2009.82) | 35480 | 35502.41 | (35352.63, 35649.46) |
| GI-8c | 2070.02 | 2069.91 | (2069.78, 2070.07) | 38220 | 38216.54 | (38058.91, 38372.02) |
| GI-9 | 2099.61 | 2099.65 | (2099.64, 2099.66) | 40160 | 40163.26 | (40001.66, 40322.67) |
| GI-10 | 2124.03 | 2124.38 | (2124.26, 2124.47) | 41460 | 41484.34 | (41320.07, 41647.05) |
| GI-11 | 2157.49 | 2159.20 | (2159.02, 2159.37) | 43340 | 43469.75 | (43300.83, 43637.19) |
| GI-12c | 2222.30 | 2222.27 | (2222.16, 2222.39) | 46860 | 46860.6 | (46685.25, 47035.35) |
| GI-13b | 2253.84 | 2254.11 | (2254.05, 2254.17) | 49120 | 49135.6 | (48957.52, 49313.66) |
| GI-13c | 2256.89 | 2257.39 | (2257.26, 2257.54) | 49280 | 49313.32 | (49134.13, 49491.73) |
| GI-14b | 2295.90 | 2296.00 | (2295.81, 2296.17) | 51660 | 51666.83 | (51482.6, 51849.78) |
| GI-14c | 2340.38 | 2340.03 | (2339.93, 2340.13) | 53960 | 53943.9 | (53753.12, 54131.19) |
| GI-14d | 2341.38 | 2341.55 | (2341.52, 2341.59) | 54020 | 54028.81 | (53837.72, 54215.99) |
| GI-14e | 2345.52 | 2345.65 | (2345.59, 2345.70) | 54220 | 54233.81 | (54042.35, 54421.9) |
| GI-15.1 | 2355.34 | 2355.35 | (2355.33, 2355.36) | 55000 | 55002.51 | (54810.67, 55190.97) |
| GI-15.2 | 2366.32 | 2366.56 | (2366.47, 2366.64) | 55800 | 55824.2 | (55630.79, 56013.95) |
| GI-16.1b | 2397.35 | 2397.36 | (2397.28, 2397.47) | 57960 | 57959.99 | (57762.56, 58153.23) |
| GI-16.1c | 2398.78 | 2398.67 | (2398.58, 2398.75) | 58040 | 58035.39 | (57838.32, 58228.46) |
| GI-16.2 | 2402.55 | 2402.30 | (2402.25, 2402.34) | 58280 | 58266.31 | (58068.78, 58459.55) |
| GI-17.1a | 2409.78 | 2409.50 | (2409.37, 2409.66) | 58780 | 58765.24 | (58567.02, 58960.27) |
| GI-17.1b | 2410.65 | 2411.26 | (2411.04, 2411.45) | 58840 | 58876.59 | (58677.75, 59072.46) |
| GI-17.1c | 2415.01 | 2414.83 | (2414.77, 2414.89) | 59080 | 59069.05 | (58870.83, 59265.27) |
| GI-17.2 | 2420.44 | 2420.70 | (2420.64, 2420.76) | 59440 | 59465.5 | (59266.65, 59662.2) |

**Table 1.** Linear ramp model fits for the NGRIP depth of 29 abrupt warming events, as well as the full dating uncertainty. Includes the depth $z^*$ and dating $y^*$ from Rasmussen et al. (2014), as well as the posterior marginal mean and 95% credible intervals for the estimated onset depth $Z^*$ and age $Y^*$.

| | Capron et al. (2021) | | | Buizert et al. (2015) | |
|---|---|---|---|---|---|
| Transition | Depth (m) | $\delta^{18}$O onset (yb2k) | $Ca^{2+}$ onset (yb2k) | Depth (m) | $\delta^{18}$O onset (yb2k) |
| GI-1e | 1604.89 | 14700 | 14708 | 1604.05 | 14 628 |
| GI-2.2 | 1794.49 | 23400 | 23428 | | |
| GI-3 | 1869.35 | 27788 | 27797 | 1869.00 | 27 728 |
| GI-4 | 1891.77 | 28911 | 28912 | 1891.27 | 28 838 |
| GI-5.2 | 1952.26 | 32528 | 32540 | 1951.66 | 32 452 |
| GI-7c | 2010.12 | 35510 | 35507 | 2009.62 | 35 437 |
| GI-8c | 2070.22 | 38231 | 38239 | 2069.88 | 38 165 |
| GI-10 | 2124.46 | 41482 | 41494 | 2123.98 | 41 408 |
| GI-11 | 2159.33 | 43471 | 43366 | 2157.58 | 43 297 |
| GI-12c | 2222.71 | 46887 | 46896 | 2221.96 | 46 794 |
| GI-14e | 2345.73 | 54235 | 54233 | 2345.39 | 54 164 |
| GI-15.1 | 2355.41 | 55006 | 55038 | 2355.17 | 54 940 |
| GI-15.2 | 2366.71 | 55831 | 55831 | 2366.15 | 55 737 |
| GI-16.2 | 2402.53 | 58279 | 58298 | 2402.25 | 59 018 |
| GI-17.1c | 2415.01 | 59080 | 59095 | 2414.82 | 59018 |
| GI-17.2 | 2420.98 | 59480 | 59483 | 2420.35 | 59 386 |

**Table 2.** Greenland interstadial transitions observed in the NGRIP record. Includes the median onset NGRIP depth and age for the $\delta^{18}$O and $Ca^{2+}$ proxy records as reported in Capron et al. (2021), and the $\delta^{18}$O record presented in Buizert et al. (2015).

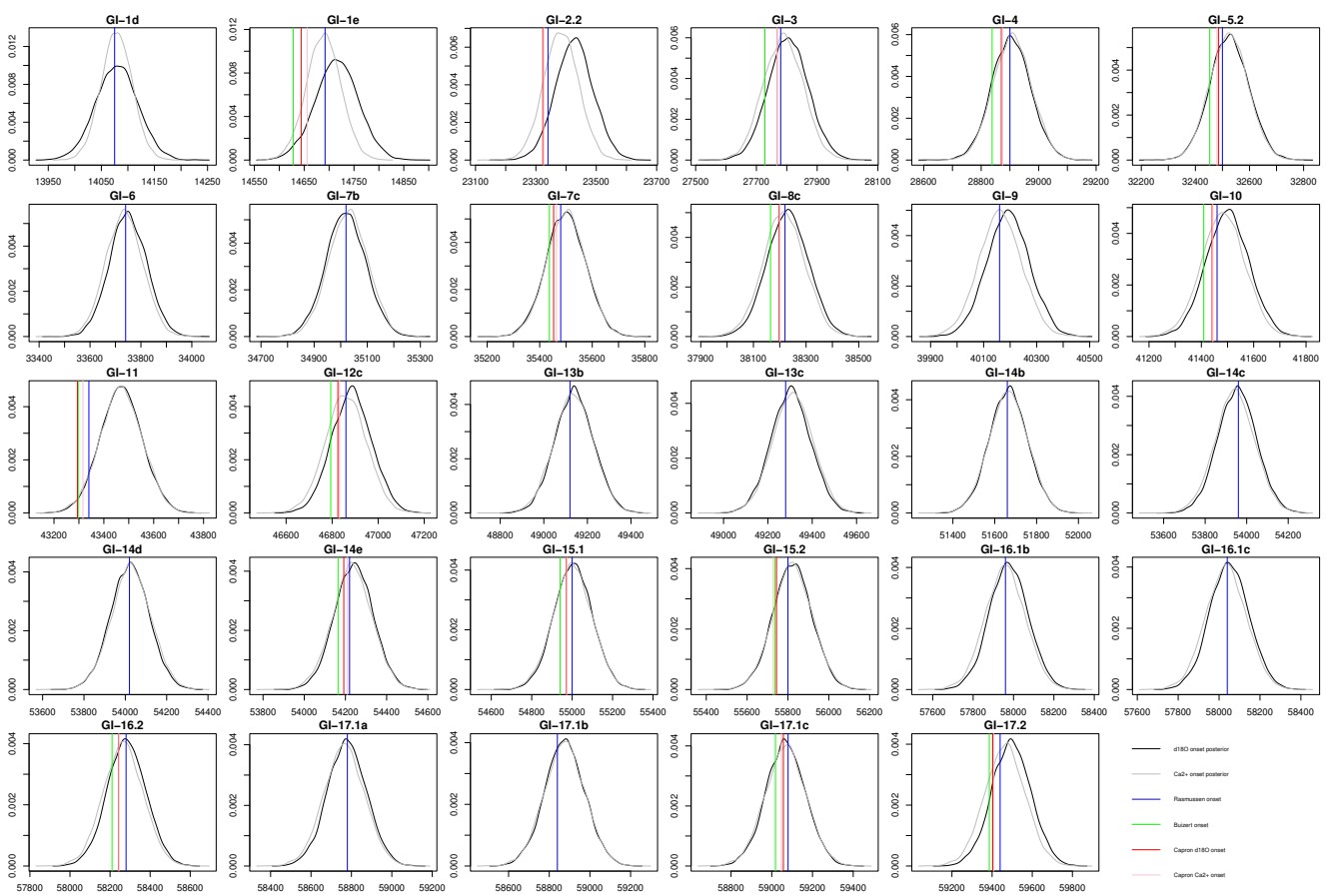

**Figure 7.** Posterior distributions of the onset ages of abrupt climate transitions for the NGRIP $\delta^{18}$O (black) and $Ca^{2+}$ (gray) proxy records. The results are compared to ages reported by Rasmussen et al. (2014) (blue), Buizert et al. (2015) (green) and Capron et al. (2021) (red and pink for the $\delta^{18}$O and $Ca^{2+}$ record respectively. The Ca2+ based posterior distributions rely on modeled chronologies which are presented in Appendix B and which are not shown in the main text

## 5 Discussion

Motivated by the relationship between air temperature on the one hand and the water-holding capacity and thus precipitation on the other hand, we assumed a linear dependency of the number of incremental layers per 5cm on respective values of $\delta^{18}O$, which serves as an air temperature proxy. The only other climate proxy variable which is available from the NGRIP ice core at the same resolution over the same time period is the $Ca^{2+}$ particle concentration (Ruth et al., 2003). From visual inspection one can already see that the negative logarithm of the $Ca^{2+}$ mass concentration record shows high covariability with the $\delta^{18}O$ record - see for example Fig. 1 of Rasmussen et al. (2014). Changes in the $Ca^{2+}$ concentrations are interpreted as changes in the local and hemispheric atmospheric circulation (e.g. Ruth et al., 2007; Schüpbach et al., 2018; Erhardt et al., 2019), which would also affect the amount of precipitation over Greenland. Above, we have focused on the $\delta^{18}O$; and analogous analysis for $-\log Ca^{2+}$ is presented in Appendix B. The results of the $-\log Ca^{2+}$-based chronology modeling are in very good agreement with the ones based on $\delta^{18}$, which corroborates our methodology.

Adopting Gaussian noise models in principle allows for a negative modelled number of incremental layers in a 5cm incremental core segment, which does not seem plausible from a physical perspective. To avoid this, for comparison one could log-transform the age increments and then apply the modeling procedure. The model output would then have to be transferred back by taking the exponential. However, it turns out that the log transformation induces systematic deviations of the mean model output age from the reported GICC05 age. With the chances of negative layer increments being fairly small (less than 5%), overall the original model outperforms the log-transformed model. A more detailed discussion on the log-transformed modeling approach is given in appendix C.

From figure 2b we observe that the variance of the residuals increases slightly with increasing depth down in the core. This heteroskedasticity can be incorporated into our latent Gaussian model by assuming that the variance depends on the core depth in some predefined way. As this implementation is rather technical we consider it beyond the scope of the current paper. Regardless, assuming constant variance appears to be a good first order approach.

## 6 Conclusions

We have developed a general statistical framework for quantifying the age-depth uncertainty of layer-counted proxy archives. In these records the age can be determined by counting annual layers that result from seasonal variations, which in turn impact the deposition process. By counting these layers one can assign time stamps to the individual proxy measurements. However, there is a non-negligible uncertainty associated with this counting process. Proper quantification of this uncertainty is important since it carries valuable information and the error propagates to further analyses, e.g. dating of climatic events, determining cause and effect between such events, and model-data comparisons. Originally, the uncertainty of the GICC05 is quantified in terms of the maximum counting error (MCE), defined as half the number of uncertain layers. However, since this method assumes that uncertain layers are either true or false, we believe this to be an overly conservative estimate, giving too high uncertainty for deeper layers.

In our approach we express the number of layers per depth increment as the sum of a structured component and a stochastic component. The structured component represents physical layer thinning, a positive temperature-precipitation feedback, and persisting trends over individual stadials and interstadials. The stochastic component takes into account the natural variability of the layer thickness and the errors made in the counting process. After fitting the structured component in a least square manner, we find the residuals to be approximately stationary, Gaussian distributed, and to exhibit short-range autocorrelation. These summary statistics motivate to employ Gaussian white noise, or an autoregressive process of first or second order, as the stochastic part of the age-depth model.

After defining the structure of the model, we estimate all model parameters simultaneously in a hierarchical Bayesian framework. The resulting joint posterior distribution on the one hand serves as a quantification of the parameter uncertainty in the model and on the other hand allows to generate chronology ensembles that reflect the uncertainty in the age-depth relationship of the NGRIP ice core. The dating uncertainties obtained from this approach are significantly smaller than the MCE. We also find that our estimates do not deviate much from the GICC05 in terms of best estimates for the dating.

Additional information that may help to further constrain the uncertainties, such as tie points obtained via cosmogenic radionuclides (Adolphi et al., 2018), will be fed into the model in future research.

One of the largest concerns regarding the layer counting is that of a potential counting bias. Such a systematic error cannot be corrected after the counting and therefore, we investigate how a potential unknown counting bias increases the uncertainty of the presented age-depth model. If such a counting bias is restricted to $\pm 4\%$ we obtain total age uncertainties comparable to the estimates based on the MCE. Finally, we apply our method to the dating of DO events. Using a Bayesian transition onset detection we are able to combine the uncertainty of the onset depth with the corresponding age uncertainty, and to give a posterior distribution that entails the complete dating uncertainty of each transition onset. We find that previous estimates of the DO onsets reported in (Rasmussen et al., 2014) are well within our estimated uncertainty ranges both in terms of depth and age. However, the dating uncertainties of the abrupt warming event onsets are all considerably smaller than the MCE, even when accounting for the additional uncertainty associated with the onset depth.

In theory, it should be possible to apply this approach to other layered proxy records as well. However, there are some requirements that need to be fulfilled for this approach to be applicable. The first condition is that a potential layer thinning can be adequately expressed by a regression model. In our results we find the residuals to follow a Gaussian process, but it should be possible for the model to be adapted such that it supports other distributions for the residuals as well. However, depending on the model, if the residuals exhibit too long memory then this could lead to the simulation procedure having an infeasibly high computational cost. Moreover, if there are many effects in the regression model there needs to be sufficient data to achieve proper inference.

*Code and data availability.* The NGRIP ice core data and GICC05 timescale (Andersen et al., 2006; Svensson et al., 2008; Rasmussen et al., 2014) is available at http://www.iceandclimate.nbi.ku.dk/data/. The code used for generating the results of this paper will be uploaded as supplementary material.

## Appendix A: Latent Gaussian model formulation

In this study we consider different Gaussian models for the noise component, including independent identically distributed (iid) and first and second order autoregressive (AR) models. These models all exhibit the Markov property, meaning there is a substantial amount of conditional independence. So-called Gaussian Markov random fields are known to work really well with the methodology of integrated nested Laplace approximations (INLAs), which will grant a substantial reduction in computational cost in obtaining full Bayesian inference. However, this requires formulating our model into a latent Gaussian model where the data, here $\mathcal{D} = (\Delta y_1, ..., \Delta y_n)^\top$, depend on a set of latent Gaussian variables $\boldsymbol{X} = (X_1, ..., X_N)^\top$ which in turn depend on hyperparameters $\boldsymbol{\theta} = (\theta_1, ..., \theta_m)^\top$. This class of models constitutes a subset of hierarchical Bayesian models and is defined in three stages.

The first stage defines the likelihood of the data and how it depends on the latent variables. For the data and models used in this study we assume a direct correspondance between an observation $y_i$ and the corresponding latent variable $X_i$, which is achieved using a Gaussian likelihood with some negligible fixed variance and mean given by the linear predictor

$$\eta_k = \mathrm{E}(\Delta y_k) = bk^2 + b_x x_k + \sum_{i=1}^{p} \psi_i(z_i; a_i, c_i) + \varepsilon_k(\boldsymbol{\theta}).$$

Here, $\boldsymbol{\beta} = (b_0, b, b_x, \{a_i\}, \{c_i\})$ are known as fixed effects, even though they are indeed stochastic variables in the Bayesian framework. The noise variables $\varepsilon_k(\boldsymbol{\theta})$ are referred to as random effects since they depend on hyperparameters $\boldsymbol{\theta}$. The hyperparameters are $\boldsymbol{\theta} = \sigma_\varepsilon$ if we assume the residuals follow an iid Gaussian process, $\boldsymbol{\theta} = (\sigma_\varepsilon, \phi)$ if they follow an AR(1) process and $\boldsymbol{\theta} = (\sigma_\varepsilon, \phi_1, \phi_2)$ if they follow an AR(2) process. All random terms in the predictor, and the predictor itself, are included in the latent field $\boldsymbol{X} = (\boldsymbol{\eta}, \boldsymbol{\beta}, \boldsymbol{\varepsilon})$. The latent field is assigned a prior distribution in what is the second stage of defining a latent Gaussian model. For latent Gaussian fields this prior is multivariate Gaussian

$$\boldsymbol{X} \mid \boldsymbol{\theta} \sim \mathcal{N}(\boldsymbol{\mu}(\boldsymbol{\theta}), \boldsymbol{\Sigma}(\boldsymbol{\theta})),$$

Specifically, we assume vague Gaussian priors for $\boldsymbol{\beta}$, while the prior for $\boldsymbol{\varepsilon}(\boldsymbol{\theta})$ is either an iid, AR(1) or AR(2) process. The predictor $\boldsymbol{\eta}$ is then a Gaussian with mean vector corresponding to the linear regression $\boldsymbol{a}(\boldsymbol{\beta})$ and covariance matrix given by the covariance structure of the assumed noise model.

The third and final stage of the latent Gaussian model definition is to specify a prior distribution on the hyperparameters. We use the default prior choices included in the R-INLA package, which means that for all models of $\varepsilon$ considered in this paper the scaling parameter $\sigma_\varepsilon^2$ is assigned a log-gamma distribution, through the transformation $\kappa = 1/\sigma_\varepsilon^2$. When the residuals follow an AR(1) distribution we assume a Gaussian prior on the additional lag-one correlation parameter using a logit transformation $\rho = \log((1+\phi)/(1-\phi))$. For the AR(2) residuals we instead assign penalisd complexity priors (Simpson et al., 2017) on the partial autocorrelations $\psi_1 = \phi_1/(1-\phi_2)$ and $\psi_2 = \phi_2$, also using a logit transformation.

Inference is obtained by computing the posterior marginal distributions

$$\pi(X_k \mid \mathcal{D}) = \int \pi(X_k \mid \boldsymbol{\theta}, \mathcal{D})\pi(\boldsymbol{\theta} \mid \mathcal{D})\mathrm{d}\boldsymbol{\theta}$$

and

$$\pi(\theta_k \mid \mathcal{D}) = \int \pi(\boldsymbol{\theta} \mid \mathcal{D}) \mathrm{d}\boldsymbol{\theta}_{-k}.$$

The notation $\theta_k$ refers to the $k$th hyperparameter, and $\boldsymbol{\theta}_{-k}$ refers to all except the $k$th hyperparameter. These integrals can be approximated efficiently using `R-INLA`, and the resulting posterior marginal distributions are included in Figure A1. The estimated hyperparameter posterior marginal means and credible intervals (given in parantheses) are $\sigma_\varepsilon^{\mathrm{iid}} \approx 0.427(0.423, 0.431)$ for iid residuals, $\sigma_\varepsilon^{\mathrm{AR(1)}} \approx 0.428(0.423, 0.434)$ and $\phi^{\mathrm{AR(1)}} \approx 0.194(0.178, 0.217)$ for AR(1) residuals, and $\sigma_\varepsilon^{\mathrm{AR(2)}} \approx 0.429(0.423, 0.438)$, $\phi_1^{\mathrm{AR(2)}} \approx 0.180(0.158, 0.210)$ and $\phi_2^{\mathrm{AR(2)}} \approx 0.108(0.089, 0.134)$ for AR(2) residuals.

## Appendix B: $Ca^{2+}$ analysis

As an alternative, we perform an analogous study using the $\log(Ca^{2+})$ as the proxy variable $x_k$, which is available at the same period and resolution (Ruth et al., 2003). Missing values in the $Ca^{2+}$ dataset are filled using linear interpolation. Performing the same analysis as described in Sect. 3 we produce joint samples of the chronologies. The resulting posterior marginal means and 95% credible intervals of the dating uncertainties are illustrated in Fig. B1, where the credible intervals obtained using an iid, AR(1) and AR(2) noise model are compared. These results are consistent with those obtained using $\delta^{18}O$ as the proxy variable, shown in Fig. 4.

## Appendix C: Log-normal distribution

A possible concern with the approach presented above is that assuming a normal distribution on the layer increments assigns a non-zero probability of negative depositions, violating monotonicity. Furthermore, one might be concerned by our choice of an additive thinning function rather than a multiplicative. Both of these issues are resolved by instead assuming a log-Normal regression model on the layer increments, i.e.

$$\log \Delta y_k = a(z_k, x_k) + \varepsilon_k, \tag{C1}$$

where $\varepsilon_k$ follows either an iid, AR(1) or AR(2) model as before. This results in the joint age variables being a multivariate process with marginals described by sums of log-normal distributions.

Simulations of the chronologies can be produced as follows. Following a latent Gaussian modeling fit, sample $a(z_k, x_k)$ and $\varepsilon_k$ using the posterior distributions. Then compute the sampled chronologies

$$y_k = y_0 + \sum_{i=1}^{k} \exp\left(a(z_i, x_i) + \varepsilon_i\right). \tag{C2}$$

The resulting dating uncertainty is illustrated in Fig. C1, and is more curved than for the original model. This suggests that the normal distribution is a better fit to the observed layer-increments. We will hence focus on the original Gaussian model in the

445 analysis of this paper.

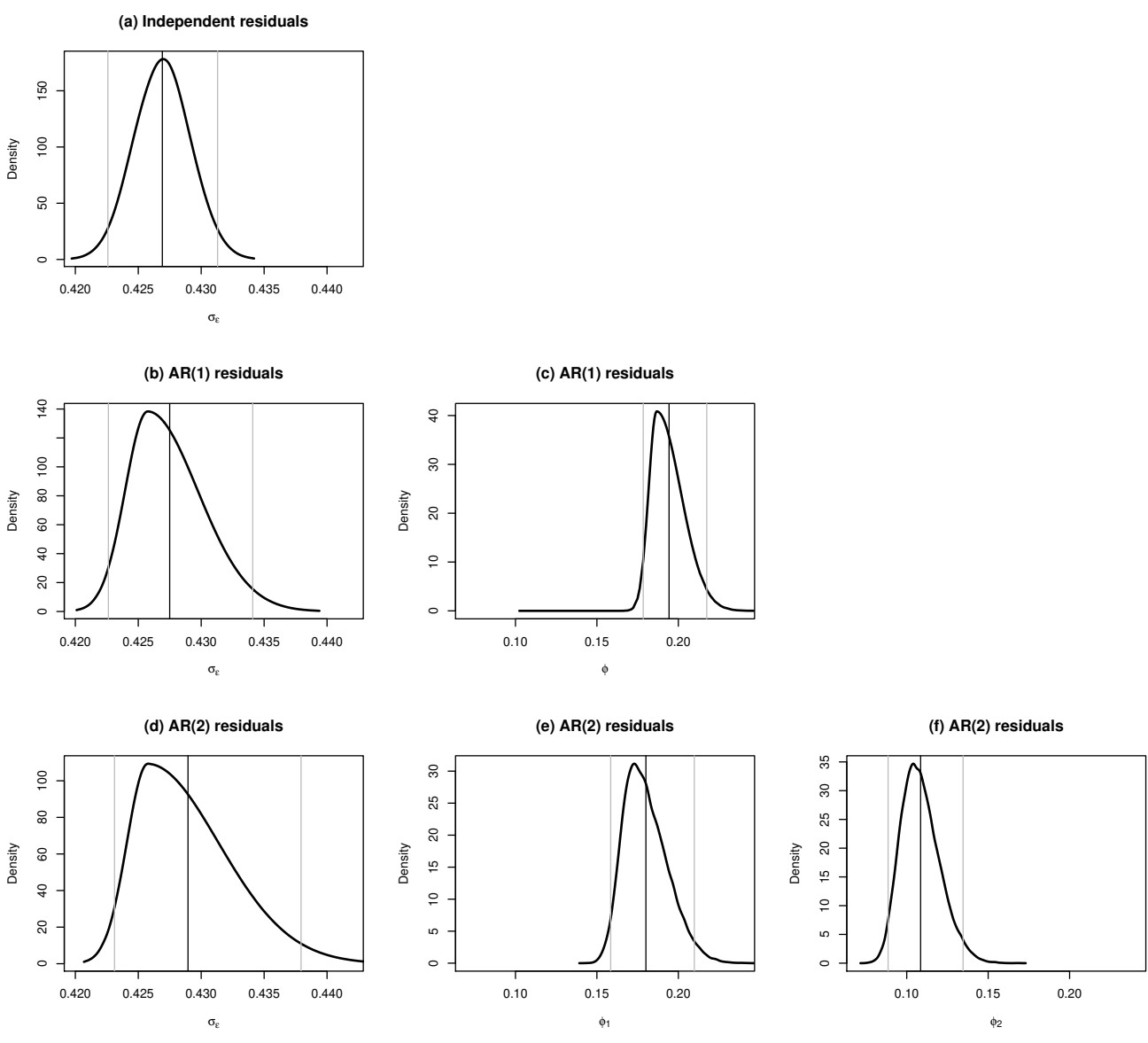

**Figure A1.** The posterior marginal distributions obtained by fitting the model with inla. Panel a shows the density of $\sigma_\varepsilon$ using iid distributed residuals. Panels b–c show the densities of $\sigma_\varepsilon$ and $\phi$ using AR(1) distributed residuals. Panels d–f show the densities of $\sigma_\varepsilon, \phi_1$ and $\phi_2$ using AR(2) distributed residuals. The vertical lines represent the mean (black) and 95% credible intervals (gray).

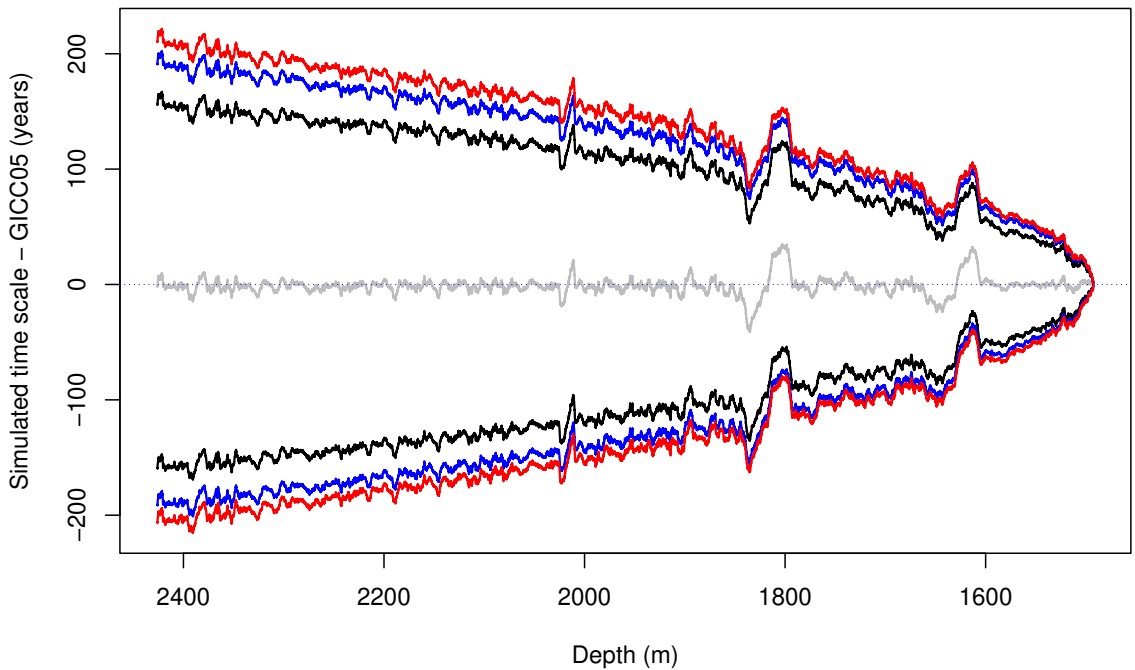

**Figure B1.** 95% credible intervals of the dating uncertainty distribution when $\log(Ca^{2+})$ is used as the proxy covariate. The GICC05 time scale has been subtracted and the noise is modeled using iid (black), AR(1) (blue) and AR(2) (red) noise models. Only the posterior marginal mean computed using AR(1) distributed noise is included (gray) since it is very similar to the mean obtained using other noise assumptions.

## Appendix D:  Determination of the fitting windows

The estimation for the onset depth is sensitive to the choice of the data window which represents the transition. It is important to select these carefully such that the data best represents a single linear ramp function while being of a sufficient size. As some DO events are located more closely to other transitions than others it is necessary to determine these data windows individually for each transition. As such there are indeed some transitions where it is difficult to determine a clear transition point, and a linear ramp model is not appropriate. These transitions will be omitted from our analysis. The reduction in computational cost granted by adopting the model for INLA allows us to perform repeated fits to determine the optimal data interval based on a given criteria. Specifically, we adjust both sides of the interval until we find the data window for which the fitted model yields the lowest amplitude of the AR(1) noise, measured by the posterior marginal mean of the standard deviation parameter.

We impose some restrictions on the domain of the optimal start and end point of our data interval. To achieve the best possible fit we want our interval to include both the onset and end point of the transition, which we suspect are located close to the NGRIP onset depth $z^*$ given by table 2 of Rasmussen et al. (2014). Unless the DO-events are located too close to adjacent transitions we assume the optimal interval always contains the points representing 1m above to 2.5 meter deeper than

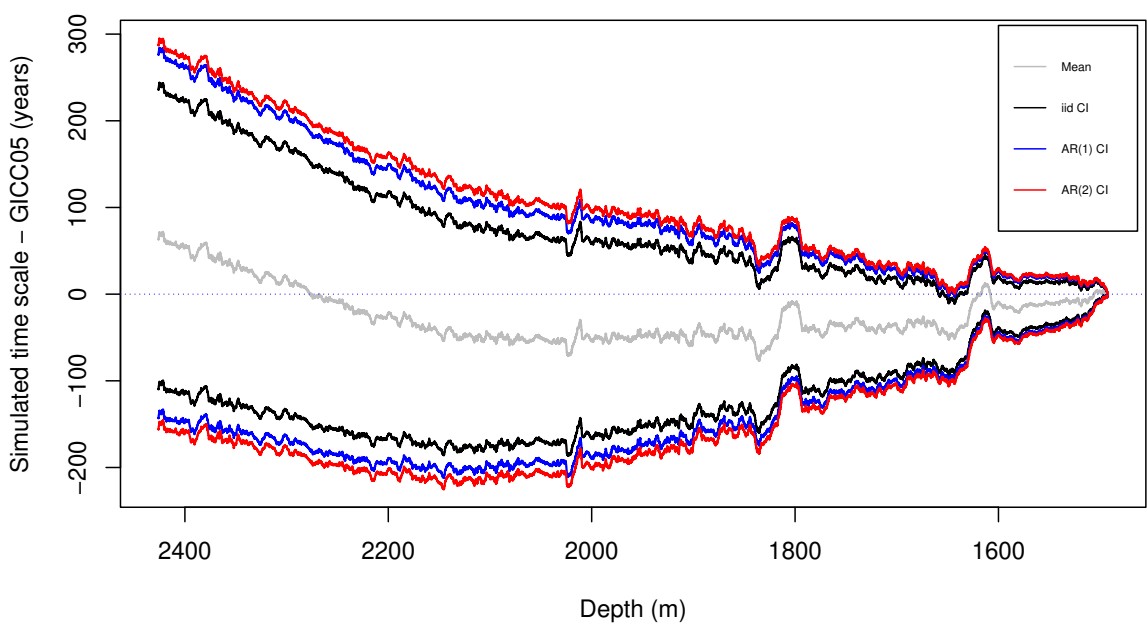

**Figure C1.** 95% credible intervals of the dating uncertainty distribution when the GICC05 time scale has been subtracted and a log-normal distribution is assumed, using iid (black), AR(1) (blue) and AR(2) (red) noise models. Only the posterior marginal mean computed using AR(1) distributed noise is included (gray) since it is very similar to the mean obtained using the other noise models.

$z^*$. These are the minimum distances from the proposed onset. Similarly, we also introduce a maximum distance from $z^*$ to

be considered for the start and end points of the data interval. This is set to be 10m above and 15m deeper than $z^*$, or at the adjacent transitions if they are located closer than this. From these intervals we create a two-dimensional grid for which we perform the INLA fit for each grid point. The start and end points of the interval representing the grid point for which INLA found the lowest noise amplitude are selected. Those events that failed to provide a decent fit after the conclusion of this procedure were discarded. This left us with 29 DO-events for which the results are displayed in table D1. Although the

GICC05 onset depths $z^*$ falls outside the 95% credible intervals for several transitions they are still remarkably close to our best estimates, considering Rasmussen et al. (2014) used a lower 20 year resolution data set.

*Author contributions.* All authors conceived and designed the study. MR conceived the idea of modeling the layer increments as a regression model, and its use for estimating dating uncertainty of DO-events. These were reworked by EMN, NB and KR, and adopted for a Bayesian framework by EMN. Further extensions to the model were conceived by EMN, NB and KR. EMN wrote the code and carried out the analysis.

EMN, KR and NB discussed the results, drew conclusions and wrote the paper with input from MR.

| Event | $k^*$ | Index interval | $z^*$ (m) | Depth Interval (m) |
|---|---|---|---|---|
| GI-1d | 1648 | (1622, 1711) | 1574.8 | (1573.5, 1577.95) |
| GI-1e | 2245 | (2175, 2285) | 1604.65 | (1601.15, 1606.65) |
| GI-2.2 | 6016 | (5986, 6079) | 1793.2 | (1791.7, 1796.35) |
| GI-3 | 7534 | (7482, 7574) | 1869.1 | (1866.5, 1871.1) |
| GI-4 | 7983 | (7929, 8023) | 1891.55 | (1888.85, 1893.55) |
| GI-5.2 | 9185 | (9106, 9223) | 1951.65 | (1947.7, 1953.55) |
| GI-6 | 9643 | (9572, 9683) | 1974.55 | (1971, 1976.55) |
| GI-7b | 10093 | (10068, 10142) | 1997.05 | (1995.8, 1999.5) |
| GI-7c | 10341 | (10169, 10404) | 2009.45 | (2000.85, 2012.6) |
| GI-8c | 11552 | (11352, 11592) | 2070 | (2060, 2072) |
| GI-9 | 12144 | (12098, 12174) | 2099.6 | (2097.3, 2101.1) |
| GI-10 | 12633 | (12563, 12690) | 2124.05 | (2120.55, 2126.9) |
| GI-11 | 13302 | (13102, 13386) | 2157.5 | (2147.5, 2161.7) |
| GI-12c | 14598 | (14417, 14718) | 2222.3 | (2213.25, 2228.3) |
| GI-13b | 15229 | (15212, 15274) | 2253.85 | (2253, 2256.1) |
| GI-13c | 15290 | (15245, 15339) | 2256.9 | (2254.65, 2259.35) |
| GI-14b | 16070 | (16054, 16110) | 2295.9 | (2295.1, 2297.9) |
| GI-14c | 16960 | (16881, 16983) | 2340.4 | (2336.45, 2341.55) |
| GI-14d | 16980 | (16968, 16992) | 2341.4 | (2340.8, 2342) |
| GI-14e | 17062 | (16986, 17185) | 2345.5 | (2341.7, 2351.65) |
| GI-15.1 | 17259 | (17247, 17272) | 2355.35 | (2354.75, 2356) |
| GI-15.2 | 17478 | (17365, 17610) | 2366.3 | (2360.65, 2372.9) |
| GI-16.1b | 18099 | (18087, 18111) | 2397.35 | (2396.75, 2397.95) |
| GI-16.1c | 18128 | (18108, 18146) | 2398.8 | (2397.8, 2399.7) |
| GI-16.2 | 18203 | (18190, 18215) | 2402.55 | (2401.9, 2403.15) |
| GI-17.1a | 18348 | (18315, 18371) | 2409.8 | (2408.15, 2410.95) |
| GI-17.1b | 18365 | (18353, 18431) | 2410.65 | (2410.05, 2413.95) |
| GI-17.1c | 18452 | (18387, 18464) | 2415 | (2411.75, 2415.6) |
| GI-17.2 | 18561 | (18531, 18608) | 2420.45 | (2418.95, 2422.8) |

**Table D1.** The optimal interval for the data window for fitting a linear ramp model to 29 DO-events, expressed in terms of depth and the corresponding index in our data. The Rasmussen et al. (2014) onset depths $z^*$ was used as a starting midpoint in the optimization procedure.

*Competing interests.*  The authors declare that they have no conflict of interest.

*Acknowledgements.*  This research is TiPES contribution #136 and has been supported by the European Union Horizon 2020 research and innovation program (grant no. 820970).

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
