# Peer review of "Code used to generate the results are available at the GitHub repository"

_Climate of the Past, 2021_

## Author Comment (AC1)

The manuscript introduces a statistically robust uncertainty estimate of the glacial section of the layer counted Greenland ice core chronology GICC05 given certain assumptions. The result is then applied to provide precise estimates of the timing and the duration of the abrupt warming transitions observed in the Greenland climate records for the 11-60 ka period. I find the study relevant and well presented, but I encourage the authors the take a step further and compare their results to results of existing studies and to make further application of their results in order to increase the relevance of the study to the paleo community. Although it appears reasonable, I have to admit that I am unable to judge the validity of the details of the statistical approach. I hope that somebody more skilled will be able to assess that part. I do have a few comments of more general character as outlined in the following.

Answer: We thank Anders Svensson for his helpful review. The comments will be addressed point-by-point below.

General comments:

It is stated repeatedly that the error estimation method provided in this study is a general method that can be directly applied to other types of layer counted records. If that is so, I suggest the authors provide examples of what other types of layer counted datasets the method could be applied for. If the method is not directly applicable to other types of records, I suggest that is reflected in the title of the manuscript. Note that some of the assumptions made along the way - eg that there exists a linear relationship between d18O and snow accumulation or that there is a linear trend in d18O within stadial or interstadial periods (lines 144-145) - are not general features that will be found in records from other types of archives (their validity can also be discussed for ice cores, but that is not the point here.)

Answer: We should clarify that the statistical framework in a broader sense is general, meaning that we can model layer-counted archives using a Bayesian regression model and an appropriate Gaussian noise term as proposed in our manuscript. The specific regression terms and parameters should of course be adjusted to each given application, and this flexibility of the model allows its underlying principles to be applied widely. Please note that we plan to build a flexible R-package based on the presented methodology, which will be flexibly applicable to different kinds of archives.

We agree that the specificity of the presented manuscript would be better reflected in a more specific title and will therefore change the title to: "Comprehensive uncertainty estimation of the timing of Greenland warmings of the Greenland Ice core records"

A comment on notation. The Dansgaard-Oeschger (DO) events are quite well defined in the Greenland ice cores records, most detailed in [Rasmussen et al., 2014] that you are citing. There are 25 major events and numerous sub-events throughout the last glacial period. You therefore risk to create confusion when you state in the manuscript (eg in the Table 1

caption and line 285) that you have identified 29 DO events in the 14-60 ka period alone. Name them 'abrupt warming transitions' or something else, but not DO-events.

Answer: We agree that the term is inaccurate and confusing. As suggested, we will change it to the more accurate term "abrupt warming transitions".

There are numerous studies in which the timing and duration of the onsets of the last-glacial abrupt warming events have been investigated (apart from [Rasmussen et al., 2014]), and I think it would be useful to compare your results to some of those. Perhaps the most obvious study to compare to is that of [Capron et al., 2021] in which the duration of the warming transitions is determined in several Greenland ice cores and for several types of records. Another study that determines the timing of the mid-point of the NGRIP d18O warming transitions is that of [Buizert et al., 2015]. Other approaches are taken in [Rousseau et al., 2017] and [Lohmann, 2019]. How does your results compare to the existing studies, for which applications is your approach superior, and why are the timing and periods different from study to study (if they are)? It would be great to visualize the timing or duration comparison in a figure somehow. Providing the exact numbers as done in Table 1 is of course important, but if you visualize the differences in a figure we will better be able to judge how important the differences are and if there are systematic offsets of some kind.

Answer: We clarify that our primary focus in this paper is to present the associated methodology for quantifying dating uncertainties. While the application to identify abrupt warming transitions and their timing are meant to be secondary, we agree that the results will certainly be of interest to readers of CP. We will add a visual presentation of the transition onsets and compare our results with the studies suggested by the referee.

In the following, I have a few suggestions for how the new findings may be applied to progress our understanding of the abrupt climate change occurring during the last glacial period. I am not expecting the authors to comment on all of the suggestions, but I think the results of the manuscript would benefit from being put into a broader context:

- Stacking of warming (or cooling) transitions. A way to investigate the general nature of the warming transitions is to stack them across warming events, similar to the approach of [Buizert et al., 2015] Figure 2b (Greenland only). If you have obtained a more precise timing of the warming transitions than in other studies, the stack of events is likely to be more abrupt in 'your stack' compared to those applying other timings of the warming transitions (following the idea of [Svensson et al., 2020] Figure 5b, where the stacking is based on two different onset determination methods)?

Answer: We appreciate the suggestion. However, we feel that the approach of stacking the events would entangle the individual uncertainties such that they are harder to control, and it is therefore not the best fit for our current statistical framework.

- Application to other Greenland ice cores. Identification of the precise timing of warming events in the NGRIP ice core is great, but what if the transitions or the timing of the transitions look different in other Greenland ice cores (this appears to be a conclusion of the [Capron et al., 2021] study)? There is no reason to think that the NGRIP isotope record is superior to that of the other deep Greenland ice cores (eg GRIP, GISP2, NEEM). All of the Greenland ice cores have been precisely synchronized by volcanic events [Rasmussen et al., 2013; Seierstad et al., 2014], so it should be possible to investigate the timing of the warming events in multiple Greenland cores or in a stack of cores (see an example of how important the differences are between cores for the onsets of the Greenland warming transitions: https://cp.copernicus.org/preprints/cp-2020-160/ Figure 2)?

Answer: These applications are indeed very interesting. As our paper is intended to be more methodological, we have not considered applying the model to other Greenland ice cores. It is, however, something we consider for future work. We also encourage others to apply the model to other ice cores. The code associated with our model will be published upon publication (or request). As the current version of the code might be rather technical in certain parts we intend to produce an accessible and flexible R-package to make the model more accessible to non-statisticians.

- Application to other data series: You are focusing on the d18O record that is indeed seen as the main ice-core climate proxy record. In [Rasmussen et al., 2014], however, the climate transitions are also identified in the Calcium record that has higher temporal resolution and may show a different climatic features. I was wondering if your method could also be applied to the NGRIP Calcium record? Or perhaps the dust record as applied in [Lohmann, 2019] and [Rousseau et al., 2017]. How does the transition onset times compare for the different records using your uncertainty estimate?

Answer: Thank you for this suggestion, an analogous analysis will be presented (in short) in a revised version of the manuscript.

- Extending the approach to the entire last glacial period. Since your approach to estimate the time scale uncertainty is not relying on the layer counting uncertainty (MCE) your method should be equally applicable to a modelled time scale? We are currently lacking precise estimates of the warming transitions for the early part of the last glacial period, so why not extend the study to cover the entire last glacial period?

Answer: The model should be applicable to the full length glacial record. In fact, by taking advantage of the Markov property present in AR(p) processes we ensure great computational efficiency that allows inference to be obtained in linear time (and memory). Obviously, the quality of the modeling depends on the available data resolution. Hence, we restricted our investigation to the time period for which the record is available consistently at 5cm resolution. We will consider applying our approach to longer, yet lower resolved time series in the future.

Specific comments:

Line 3: stadial and interstadial periods may not be known to all readers.

Line 67: 'up' or 'down' to depth z?

Line 125: I guess snow can both be removed and added by wind? Sublimation is another factor. Maybe not necessary to introduce those effects here.

Line 301 onwards: There are several reasons why the transition depths derived in [Rasmussen et al., 2014] may differ from those of the present study other than that they are obtained in 20 yr resolution from visual inspection. Rasmussen et al., identify the transitions in three Greenland ice cores using two different proxies (d18O and Calcium), whereas you determine the transitions in a single record from a single core. For a direct comparison, you should - in principle - derive the transitions from the same six records.

Figure 6, caption: in the last line there is mentioning of a red vertical line that has not shown up in my version of the figure.

In Table 1 there seems to be an error for the Z* CI column where all numbers are identical?

Line 314: You are describing a general framework, but not all paleo-archives apply d18O as a climate proxy.

Answer: The specific comments will be addressed in the revised manuscript.

---

## Author Comment (AC2)

In this article, the authors propose a the structure of the age density in the layer-counted GICC05 chronology for NorthGRIP.

They model the structure of the age density (the number of layer per depth unit) as the sum of a 2nd-order polynomial term (representing the layer thinning due to ice flow), a d18O-related term, and a depth-related term that is unique to each stadial or interstadial interval.

The rest of the residuals is described as noise, which is best characterized by a AR(1) or AR(2) process.

It is then discuss whether there could be a systematic bias in the Maximum Counting Error (MCE), but no clear conclusion is drawn in this section.

Later, in an application section, the depths of the DO transitions are determined using a statistical framework, and the age uncertainties are derived from the depth uncertainties and from the age uncertainties previously derived.

The paper is well written, and the application of statistical tools is rigorous, as far as I could understand, but I have a few comments which could help to make the paper more relevant.

Answer: We thank Frédéric Parrenin for his helpful review. The comments will be addressed point-by-point below.

General comments:

1. The title is misleading, since here you really focus on NorthGRIP/GICC05, with its particular uncertainty structure. I would therefore use a more specific title. Moreover, the part related to the determination of the DO transitions is not mentionned in the title, while it is an interesting application.

Answer: We agree with both reviewers that the application of determining the abrupt warming transitions should receive more emphasis. We will change the title to "Comprehensive uncertainty estimation of the timing of Greenland warmings of the Greenland Ice core records"

2. The paper pretend to model the GICC05 uncertainty, but I think it rather models the GICC05 age density. It is the first interest of this paper, to try to explain the GICC05 age density as far as possible with mathematical regressions, and to have an as small as possible residual term.

Answer: We think that a probabilistic age-depth model fitted to empirical age densities is an appropriate description of the age uncertainty. Similar interpretations can be found in other uncertainty frameworks such as the BACON (Blaauw and Christen 2011) where a first order

autoregressive gamma distribution is proposed for the accumulation rates (though without a regression component) and fitted to the data using a Markov Chain Monte Carlo approach.

3. The modeling of the thinning process as an additive 2nd-order polynome is questionable. First, the thinning function is not additive, but rather multiplicative. The analysis should therefore be applied to the log of the age density, so that multiplicative terms become additive terms. Second, there are more appropriate formulation of the thinning function, the like Lliboutry profile (see for example Parrenin et al., The Cryosphere, 2017). Although here I am not sure it will make a big difference, since the difference with a 2nd order polynome is important only in the very bottom section.

Answer: We agree that it is more suitable to apply our regression to the logarithm of the layer increments instead. Moreover, this would also ensure monotonicity by not allowing negative layers, which makes the model more realistic. We have conducted a preliminary analysis which seems to suggest that the residuals are more skewed and have a more apparent degree of heteroskedasticity. We have been able to find a way to model non-constant variance within our existing framework, but this might be too technical to be included in this paper. We are currently working on another paper related to this age-depth model that we aim to publish in a statistical journal where this might be a better fit. We will provide a comment on this in the revised manuscript.

4. The depth-related term in the age density is an interesting observation, but there is no physical explanation for it. It could be interesting to discuss some hypotheses.

Answer: We are not entirely sure if we understand this comment correctly. If the referee refers to the second order polynomial fit as the 'depth-related term' , then a physical explanation of the depth related term is included in lines 141-143.

5. The modeling of the stochastic residuals with AR(1) or AR(2) processes is questionable, since the age density is calculated every 5 cm (if I understood correctly). Therefore, an AR(1) process does not represent the same time memory in the top or bottom parts of the records.

Answer: Similar to other frameworks for uncertainty quantification, we do not model the actual climate process, but merely the thinning of the ice core. Hence the AR processes describe the interdependencies of different ice core slices and not climatic memory (for which we agree that a temporal axis would be more appropriate). Other uncertainty quantification frameworks treat memory similarly. A depth-varying memory would be possible to incorporate, and could possibly be addressed in future work.

6. The residuals are described as a gaussian process, but it seems from Fig. 2b that the standard deviation is not constant but rather depth dependent. This is not really discussed as far as I understood.

Answer: We agree that there is an apparent depth dependency in the residuals' amplitudes. However, the amplitude modulation is sufficiently weak, such that assuming a homoscedastic gaussian noise process is still a reasonable first order modeling approach. As already mentioned, in upcoming research we will incorporate the log transformation as proposed by the referee, which simultaneously will allow for better control of the heteroscedasticity in the residuals.

7. Regarding biases, when I read the abstract I got interested because I thought that such bias would be estimated. This could be the case by using a more accurate (in absolute ages) and independent chronology, like the U-Th dating of the Hulu cave speleothem record. But this is not the case here. I don't really see which message we get from this section on the biases, since there is no clear quantification at the end.

Answer: Knowledge about the counting process and underlying biases can be readily incorporated into our method if such information is available. We just give some examples of possible biases and demonstrate how such biases could affect the overall dating uncertainty. In a revised manuscript we will modify the abstract and will write:

"We show how the effect of a potential counting bias can be incorporated in our framework. Furthermore we present refined estimates of the occurrence times of Dansgaard-Oeschger events evidenced in Greenland ice cores together with a complete uncertainty quantification of these timings." instead of "We show how the effect of an unknown counting bias can be incorporated in our framework and present refined estimates of the occurrence times of Dansgaard-Oeschger events evidenced in Greenland ice cores together with a complete uncertainty quantification of these timings."

We hope that this change makes clear that in this study we do not intend to estimate the counting bias explicitly.

In principle, a bias can be estimated by synchronizing our timescale to tie points obtained from other archives. Synchronization with other archives requires additional, lengthy technical derivations, and will therefore be addressed in an individual paper on how to incorporate tie-points into this framework, which we hope to submit soon.

8. Regarding the identification of the DO transitions, it only appears as an application of the uncertainty quantification method, while I agree with Anders Svensson it has an strong interest by itself, in particular for the stacking of these transitions. Maybe giving more focus on this aspect could make the paper more relevant. I also agree with Anders Svensson on the interest of applying this method to other Greenland cores, other datasets or older time periods.

Answer: Thank you! We will in the revision put more emphasis on identifying the abrupt warming transitions, as we agree it will be of interest to the readers of the journal. However, the main aim of this paper remains to present the methodology. As such, we have, in an

effort to avoid overloading this paper, only applied it to one ice core. We will, however, consider applying the model to other archives in future work.

Specific comments:

l. 245: "of the GICC05 chronology"

l. 246 "over counted or missed." (missing dot)

fig. 6 legend, 2nd line: "linear ramp"

Answer: The specific comments will be addressed in the revised manuscript.